# The Derlin-1-Stat5b axis maintains homeostasis of adult hippocampal neurogenesis

Naoya Murao [ID][1], Taito Matsuda[2], Hisae Kadowaki [ID][1], Yosuke Matsushita[3,4], Kousuke Tanimoto[5], Toyomasa Katagiri[3,4], Kinichi Nakashima [ID][2✉] & Hideki Nishitoh [ID][1,6✉]

## Abstract

Adult neural stem cells (NSCs) in the hippocampal dentate gyrus continuously proliferate and generate new neurons throughout life. Although various functions of organelles are closely related to the regulation of adult neurogenesis, the role of endoplasmic reticulum (ER)-related molecules in this process remains largely unexplored. Here we show that Derlin-1, an ER-associated degradation component, spatiotemporally maintains adult hippocampal neurogenesis through a mechanism distinct from its established role as an ER quality controller. Derlin-1 deficiency in the mouse central nervous system leads to the ectopic localization of newborn neurons and impairs NSC transition from active to quiescent states, resulting in early depletion of hippocampal NSCs. As a result, Derlin-1-deficient mice exhibit phenotypes of increased seizure susceptibility and cognitive dysfunction. Reduced Stat5b expression is responsible for adult neurogenesis defects in Derlin-1-deficient NSCs. Inhibition of histone deacetylase activity effectively induces Stat5b expression and restores abnormal adult neurogenesis, resulting in improved seizure susceptibility and cognitive dysfunction in Derlin-1-deficient mice. Our findings indicate that the Derlin-1-Stat5b axis is indispensable for the homeostasis of adult hippocampal neurogenesis.

**Keywords** Endoplasmic Reticulum; Neural Stem Cell; Adult Neurogenesis; Seizure; 4-Phenylbutyric Acid
**Subject Category** Neuroscience

## Introduction

The adult mammalian brain retains neural stem/precursor cells (NS/PCs) in restricted brain regions such as the subventricular zone (SVZ) of the lateral ventricle and the subgranular zone (SGZ) of the hippocampal dentate gyrus (DG), and these NS/PCs continuously generate neurons throughout the life of the individual (Eriksson et al, 1998; Goncalves et al, 2016). Persistent generation of neurons in the adult brain is commonly referred to as adult neurogenesis. Particularly in the DG, it plays an important role in learning and memory formation. Adult neurogenesis is disrupted in several neurological diseases associated with memory impairment (e.g., seizures, depression, schizophrenia, and Alzheimer's disease) (Kang et al, 2016; Snyder et al, 2011; Terreros-Roncal et al, 2021). Radial neural stem cells (NSCs), the source of functional neurons, are reversibly regulated to be either quiescent or proliferative (activated) by the interplay of neurogenic niche–derived signaling pathways, and this regulation of NSCs is essential for persistent neurogenesis throughout life (Bond et al, 2015; Urban et al, 2019). Furthermore, during the process of adult neurogenesis, the correct migration of newborn neurons in the adult DG is critical for physiological hippocampal function, and the mislocalization of these cells often leads to neurological dysfunction, probably due to abnormal neuronal circuit formation (Lybrand et al, 2021; Scharfman and Pierce, 2012). Recent studies have focused on the role of organelles such as mitochondria and lysosomes, as well as developmental signaling, transcriptional, and epigenetic pathways, in the regulation of neurogenesis (Beckervordersandforth et al, 2017; Kobayashi et al, 2019; Murao et al, 2016; Petrelli et al, 2023). However, the underlying mechanisms of the regulation of adult neurogenesis by organelles and related molecules are not yet fully understood. The endoplasmic reticulum (ER) is a crucial organelle involved in the regulation of lipid and glucose metabolism, $Ca^{2+}$ signaling, and proteostasis, and its quality control system is regulated by the unfolded protein response (UPR). The UPR mediates the proper folding or degradation of unfolded proteins and attenuates translation to inhibit the further accumulation of proteins in the ER. Previous studies have shown that impaired ER quality contributes to the onset and exacerbation of several neurological diseases featuring learning and memory deficits (Ghemrawi and Khair, 2020; Hetz and Saxena, 2017). Furthermore, it has been shown that the UPR or lipogenic ER stress in NSCs is closely related to cortical development and adult neurogenesis (Bowers et al, 2020; Laguesse et al, 2015). Therefore, ER function or related factors and adult neurogenesis are thought to be closely related to the mechanisms of cognitive function and neurological diseases, whereas the physiological mechanisms of these relationships remain to be elucidated.

[1]Laboratory of Biochemistry and Molecular Biology, Department of Medical Sciences, University of Miyazaki, Miyazaki, Japan. [2]Department of Stem Cell Biology and Medicine, Graduate School of Medical Sciences, Kyushu University, Fukuoka, Japan. [3]Division of Genome Medicine, Tokushima University, Tokushima, Japan. [4]National Institutes of Biomedical Innovation, Health and Nutrition, Osaka, Japan. [5]High-risk Infectious Disease Control, Graduate School of Medical and Dental Sciences, Tokyo Medical and Dental University, Tokyo, Japan. [6]Frontier Science Research Center, University of Miyazaki, Miyazaki, Japan. ✉E-mail: nakashima.kinichi.718@m.kyushu-u.ac.jp; nishitoh@med.miyazaki-u.ac.jp

An ER membrane protein, Derlin-1, mediates ER-associated degradation (ERAD) and ER stress-induced pre-emptive quality control (ERpQC), and is essential for ER quality control in general (Kadowaki et al, 2015; Kadowaki et al, 2018; Lilley and Ploegh, 2004; Ye et al, 2004). We have previously shown that the interaction of Derlin-1 with amyotrophic lateral sclerosis (ALS)-associated superoxide dismutase 1 (SOD1) mutants leads to pathological UPR and motor neuron dysfunction (Nishitoh et al, 2008). Furthermore, loss of Derlin-1 in the central nervous system (CNS) induces brain atrophy and motor dysfunction by impairing neuronal cholesterol biosynthesis, which is regulated on the ER membrane (Sugiyama et al, 2021). Chemical chaperones such as 4-phenylbutyric acid (4-PBA) can rescue the above phenotypes in Derlin-1-deficient mice, indicating that Derlin-1-mediated ER quality control is essential for brain development and function (Sugiyama et al, 2022). Derlin-1 is expressed in adult hippocampal NSCs, and its expression is high in quiescent NSCs and decreases along with the progression of adult NSC developmental stages (Shin et al, 2015). Considering these findings, we hypothesize that Derlin-1 in NSCs may be necessary for the regulation of adult hippocampal neurogenesis and related behaviors.

Here, we show that Derlin-1 is responsible for the regulation of adult hippocampal neurogenesis in a spatiotemporal manner, i.e., the transition of NSCs from active to quiescent states and the localization and survival of newborn neurons, and that NSCs are depleted early in mice with CNS-specific Derlin-1 deficiency. Furthermore, the loss of Derlin-1 (Derl1) in mice increases seizure susceptibility and impairs cognitive function. The signal transducer and activator of transcription 5b (Stat5b) was identified as a regulator of adult neurogenesis downstream of Derlin-1. Surprisingly, 4-PBA rescues the phenotype of Derlin-1-deficient mice via its inhibitory action on histone deacetylase (HDAC), not its chaperone activity. Overall, our work demonstrates that the Derlin-1-Stat5b axis is essential for the homeostasis of adult hippocampal neurogenesis and consequently plays an important role in regulating seizure susceptibility and cognitive function.

## Results

### Abnormal adult hippocampal neurogenesis due to Derlin-1 deficiency

Impaired ER function contributes to the pathogenesis of several neurological diseases characterized by cognitive dysfunction, and Derlin-1 expression in the mouse hippocampus varies across adult NSCs of different stages (Ghemrawi and Khair, 2020; Hetz and Saxena, 2017; Shin et al, 2015). To understand the role of Derlin-1 in adult neurogenesis, we analyzed mice with CNS-specific Derlin-1 deficiency (Derl1$^{NesCre}$ mice) generated by mating Derl1$^{flox/flox}$ (Derl1$^{f/f}$) mice harboring a floxed Derl1 gene with transgenic mice expressing Cre recombinase under the control of the nestin promoter [Tg(Nes-Cre)1Kag mice] (Isaka et al, 1999; Sugiyama et al, 2021). Derlin-1 protein was barely detectable in the hippocampal region or DG of Derl1$^{NesCre}$ mice (Fig. EV1A). Using DNA microarray data, we confirmed the induction of ER stress in the DG of Derl1$^{NesCre}$ mice at 2 months of age via gene set enrichment analysis (GSEA) focused on the term "response to ER stress" (Fig. EV1B), consistent with our previous findings that ER

stress is induced in the cerebellum of this mouse line (Sugiyama et al, 2021). In the mouse DG and granular cell layer (GCL), structures are fully formed at approximately 2 weeks of age, and the number of neuronal progenitor cells in the molecular layer (ML) decreases as DG development progresses (Noguchi et al, 2016). We examined whether Derlin-1 deficiency affects hippocampal development by observing GCL morphology and the number and localization of cells positive for the neuronal progenitor marker Tbr2 and the proliferating cell marker Ki67 (Tbr2$^+$ and Ki67$^+$) at postnatal day (P) 14 (Fig. EV1C). Cell proliferation during DG development was analyzed by a short pulse-label experiment with bromodeoxyuridine (BrdU) at P14 (Fig. EV1H,I). The number of Ki67$^+$ or BrdU$^+$ proliferating cells and the number and ratio of Tbr2$^+$ neural progenitor cells in the DG and ML were unchanged (Fig. EV1D–G,J), suggesting that DG development is not affected by Derlin-1 deficiency. We previously reported no difference in brain size or weight at age P0 (Sugiyama et al, 2021). Additionally, there was no clear difference in brain weight or cortical morphology according to the staining for neurons (NeuN) at P14 (Fig. EV1K,L). Collectively, these results suggest that substantial anomalies in brain development do not appear until at least P14. Next, we injected 8-week-old Derl1$^{f/f}$ or Derl1$^{NesCre}$ mice with BrdU once a day for 7 days to examine the effects of Derlin-1 deficiency on adult hippocampal neurogenesis (Fig. 1A). The numbers of BrdU$^+$ proliferating cells and BrdU$^+$ and DCX$^+$ newborn neurons were increased in Derl1$^{NesCre}$ mice (Fig. 1B–D), suggesting that Derlin-1-deficient NSCs proliferate excessively within a time frame of 1 week in the adult hippocampus.

The Derlin family consists of Derlin-1, 2, and 3. Derlin-1 and 2 are ubiquitously expressed, including in brain, while Derlin-3 is not, and both Derlin-1 and Derlin-2 play important roles in brain development and function by maintaining ER quality (Dougan et al, 2011; Nishitoh et al, 2008; Sugiyama et al, 2022). We generated Derl2$^{NesCre}$ mice and injected BrdU once a day for 7 days in 8-week-old Derl2$^{NesCre}$ and control (Derl2$^{f/f}$) mice (Fig. EV1M). The expression of ER stress–responsive genes was significantly increased in the DG of Derl2$^{NesCre}$ mice (Fig. EV1N); however, surprisingly, there was no significant difference in the number of BrdU$^+$ cells or BrdU- and DCX-double-positive cells (Fig. EV1O–Q). These results suggest that the disturbance of adult neurogenesis in Derl1$^{NesCre}$ mice might be independent of ER stress throughout the DG. We further examined whether Derlin-1 expression in neurons is involved in adult neurogenesis. Derl1$^{CaMKIIαCre}$ mice, in which Derl1 is specifically deleted in neurons, showed no changes in cell proliferation or DCX$^+$ neuron production (Fig. EV1M,R–T). We next focused on the number and location of newborn neurons in Derl1$^{NesCre}$ mice, as the number and location of newborn neurons are important in the developmental process of adult hippocampal neurogenesis. In the DG of Derl1$^{NesCre}$ mice, the numbers of both immature neurons positive for DCX (Fig. 1E,F) and mature neurons positive for the granular cell marker Prox1 (Fig. 1H,I) located in the hilus were increased. In contrast, the number of DCX$^+$ cells in the SGZ did not change (Fig. 1G). Collectively, these findings suggest that the loss of Derl1 induces ectopic neurogenesis. To examine whether the abnormally generated neurons matured, Derl1$^{NesCre}$ and control mice were analyzed 3 weeks after 7 days of BrdU injection (Fig. 1J). BrdU$^+$ and NeuN$^+$ mature neurons were not increased (Fig. 1K,L) and the survival ratio of newborn neurons was significantly decreased in the GCL of Derl1$^{NesCre}$ mice (Fig. 1M).

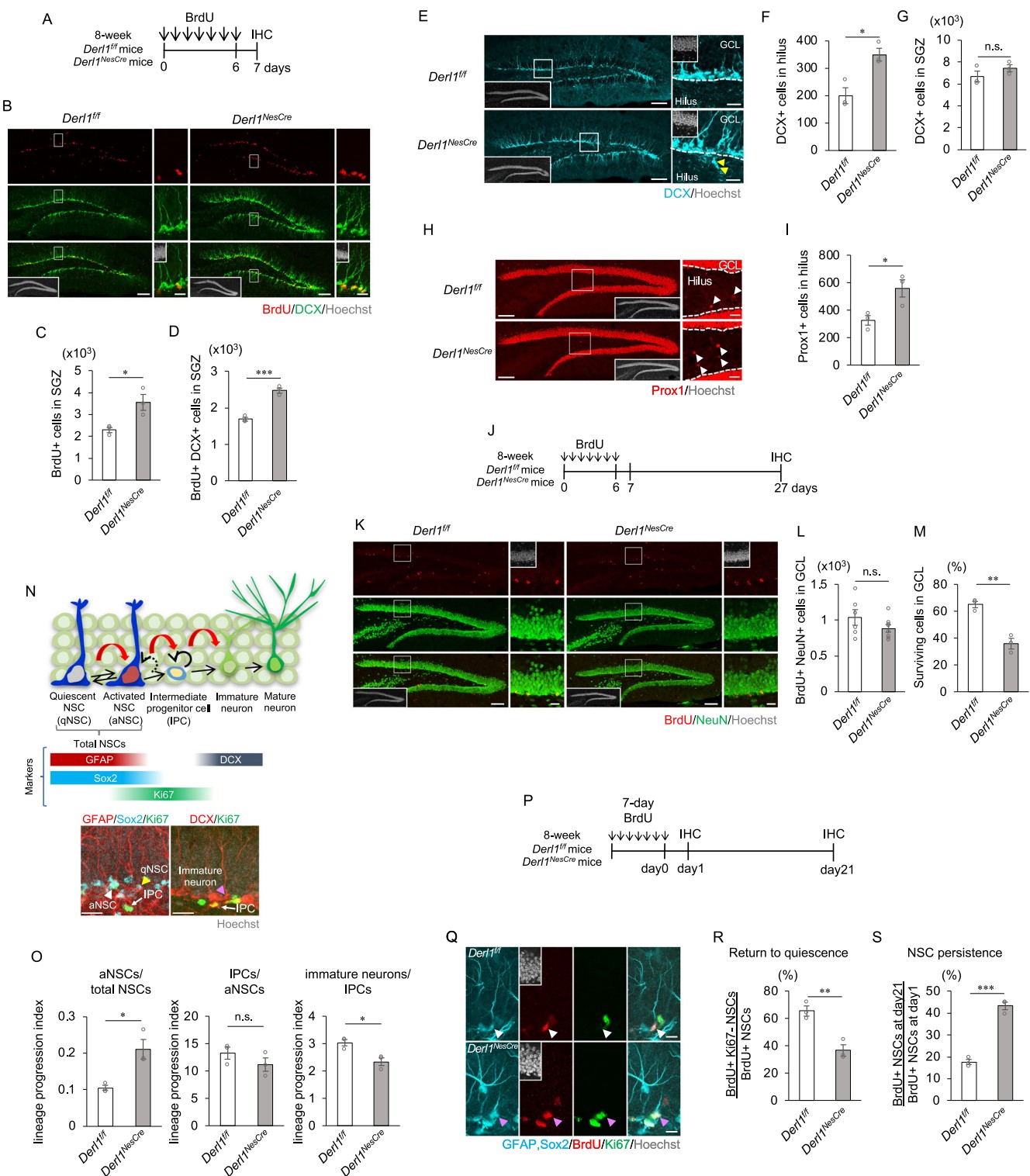

Since niche changes due to increased glial cells surrounding NSCs have a profound negative effect on the production of newborn neurons in adult hippocampal neurogenesis (Casse et al, 2018; Cope and Gould, 2019), we examined the change in glial cells in *Derl1^{NesCre}* mice. There was no difference in the number of GFAP+ and S100β+ astrocytes or Iba1+ microglia in the ML or hilus

of the DG region at 4 weeks of age (Fig. EV1U–Y). A significant difference in the number of BrdU+ S100β+ (newly generated) or active caspase3+ S100β+ (dying) astrocytes was not observed (Fig. EV1Z–AE). Taken together, these findings suggest that Derlin-1 deficiency does not affect the number of astrocytes or microglia in the DG or affect astrogenesis derived from adult

**Figure 1.   Loss of *Derl1* perturbs adult hippocampal neurogenesis.**

(A) Experimental scheme for investigating the proliferation of NS/PCs and neurogenesis in *Derl1^f/f* and *Derl1^NesCre* mice. (B) Representative immunofluorescence images of DG staining for BrdU (red), DCX (green), and Hoechst (gray; insets). The areas outlined by a white rectangle are enlarged to the right. Scale bars, 100 μm (left images) and 20 μm (right images). (C, D) Quantification of the numbers of BrdU-positive (BrdU^+) proliferating cells (C) and BrdU^+ DCX^+ newborn immature neurons (D) in the SGZ of *Derl1^f/f* and *Derl1^NesCre* mice (n = 3 mice). (E) Representative immunofluorescence images of DG staining for DCX (cyan) and Hoechst (gray; insets). The areas outlined by a white rectangle are enlarged to the right. The yellow arrowheads indicate DCX^+ ectopic immature neurons in the hilus and the dashed white lines indicate the boundaries between the GCL and the hilus. Scale bars, 100 μm (left images) and 20 μm (right images). (F, G) Quantification of the number of DCX^+ cells in the hilus (F) and SGZ (G) of *Derl1^f/f* and *Derl1^NesCre* mice (n = 3 mice). (H) Representative immunofluorescence images of the DG with Prox1 (red) and Hoechst staining (gray; insets). The areas outlined by a white rectangle are enlarged to the right. The white arrowheads indicate Prox1^+ ectopic neurons in the hilus, and the dashed white lines indicate the boundaries between the GCL and the hilus. Scale bars, 100 μm (left images) and 20 μm (right images). (I) Quantification of the number of Prox1^+ cells in the hilus of *Derl1^f/f* and *Derl1^NesCre* mice (n = 3 mice). (J) Experimental scheme for assessing neurogenesis in the DG of *Derl1^f/f* and *Derl1^NesCre* mice. (K) Representative immunofluorescence images of the DG with BrdU (red), NeuN (green), and Hoechst staining (gray; insets). The areas outlined by a white rectangle are enlarged to the right. Scale bars, 100 μm (left images) and 20 μm (right images). (L) Quantification of the number of BrdU^+ NeuN^+ newborn mature neurons in the GCL of *Derl1^f/f* and *Derl1^NesCre* mice (n = 6; *Derl1^f/f* mice, n = 8; *Derl1^NesCre* mice). (M) The percentage of BrdU^+ cells surviving between 1 and 3 weeks in the GCL of *Derl1^f/f* and *Derl1^NesCre* mice. The survival ratio was obtained by dividing the total number of BrdU^+ cells at 3 weeks (day 27 overall) by the total number at 1 day (day 7 overall) after the last BrdU injection (n = 3 mice). (N) Schematic diagram of the stage of adult neurogenesis (top panel) and specific marker proteins for each stage (middle panel). Representative immunofluorescence images of radial GFAP^+ Sox2^+ Ki67^- (qNSC), radial GFAP^+ Sox2^+ Ki67^+ (aNSC), Ki67^+ [radial GFAP^+ Sox2^+]^- (IPC), and DCX^+ Ki67^- (immature neuron) staining (bottom panel). The white arrowhead indicates representative aNSC, the yellow arrowhead indicates representative qNSC, the pink arrowhead indicates representative immature neuron, and the white allows indicate IPC. Scale bars: 20 μm. (O) Comparison of the lineage progression index between *Derl1^f/f* and *Derl1^NesCre* mice. The lineage progression index is calculated by dividing the number of cells at a defined developmental stage by the number of cells at the preceding developmental stage (aNSCs normalized to total NSCs, IPCs normalized to aNSCs, immature neurons normalized to IPCs) (n = 3 mice). (P) Experimental scheme for assessing return to quiescence of NSCs and NSC persistence in the *Derl1^f/f* and *Derl1^NesCre* mice. (Q) Representative immunofluorescence images of the SGZ with GFAP, Sox2 (cyan), BrdU (red), Ki67 (green), and Hoechst staining (gray; insets). The white arrowheads indicate returned quiescent NSC and the pink arrowheads indicate active NSC. Scale bars: 20 μm. (R) Quantification of return to quiescence of NSCs in the DG of *Derl1^f/f* and *Derl1^NesCre* mice (n = 3 mice). (S) Quantification of NSC persistence in the DG of *Derl1^f/f* and *Derl1^NesCre* mice (n = 3 mice). Bar graphs are presented as the mean ± SEM. *P < 0.05, **P < 0.01, and ***P < 0.001 by Student's t-test. n.s. not significant. Source data are available online for this figure.

hippocampal NSCs. Therefore, changes in glial cells surrounding NSCs are unlikely to affect adult neurogenesis in *Derl1^NesCre* mice.

To identify abnormalities in the stage of adult neurogenesis in *Derl1^NesCre* mice, we investigated the lineage progression index at each developmental stage using cellular markers at specific developmental stages. We first identified NSCs by staining with GFAP and Sox2 antibodies. There was no difference in the number of cells between radial Nestin^+ and Sox2^+ cells or radial GFAP^+ and Sox2^+ cells (Fig. EV1AF,AG), confirming the presence of Sox2^+ cell bodies in the SGZ and the radial process of GFAP^+ radial glial extension (radial GFAP^+). Radial GFAP^+, Sox2^+, and Ki67^+ cells were defined as activated NSCs; Ki67^+ cells minus radial GFAP^+, Sox2^+ cells, as intermediate progenitor cells (IPCs); and DCX^+ and Ki67-negative (Ki67^-) cells, as immature neurons (Fig. 1N). We calculated the lineage progression index by dividing the number of cells in a defined developmental stage by the number of cells in the preceding developmental stage. Intriguingly, we found that the lineage progression indices of activated NSCs but not of IPCs were increased in the *Derl1^NesCre* mice (Fig. 1O). In contrast, the lineage progression index of immature neurons was lower in *Derl1^NesCre* mice than in *Derl1^f/f* mice (Fig. 1O). However, the mechanism underlying the altered lineage progression index of immature neurons remains to be elucidated.

In the adult hippocampal DG, activated NSCs are known to return to a quiescent state at a certain rate (Harris et al, 2021). Because the percentage of activated NSCs was greater in *Derl1^NesCre* mice than in control mice (Fig. 1O), activated Derlin-1-deficient NSCs may be less likely to return to a quiescent state. To test this hypothesis, we investigated the rate at which activated NSCs returned to the quiescent state in vivo (Fig. 1P–R). The percentage of NSCs that returned to quiescence in the DG was markedly lower in the *Derl1^NesCre* mice than in the control mice (Fig. 1R), suggesting that Derlin-1 mediates the transition of NSCs from the active state to quiescent states. The maintenance of activated Derlin-1-deficient

NSCs was assessed by dividing the number of BrdU^+ NSCs on day 21 after the last BrdU dose by the number on day 1 and examining the ratio (Fig. 1P). The percentage of activated NSCs in the *Derl1^NesCre* mice was still greater than that in the control mice after 3 weeks (Fig. 1P,S), suggesting that activated Derlin-1-deficient NSCs are maintained over time. Taken together, these results suggest that Derlin-1 primarily regulates the quiescent and active states of hippocampal NSCs in the adult mouse brain and is also involved in the localization and survival of newborn neurons.

## Enhanced seizure susceptibility due to Derlin-1 deficiency

Ectopic localization of neurons in the hilus of the DG is frequently observed in patients with temporal lobe epilepsy, the most common form of epilepsy in adults, and in animal models of this disease (Cho et al, 2015; Hester and Danzer, 2013; Matsuda et al, 2015; Parent et al, 2006). These ectopic neurons are more excitable than neurons in the GCL (Cameron et al, 2011; Zhan et al, 2010). Having observed that the number of Prox1^+ neurons in the hilus is greater in *Derl1^NesCre* mice than in control mice, we hypothesized that *Derl1^NesCre* mice might exhibit increased seizure susceptibility. First, to examine whether *Derl1^NesCre* mice exhibit spontaneous seizure symptoms, we videotaped them continuously for 21 h in a stimulus-free, freely active state; no seizure symptoms were observed (https://zenodo.org/records/10548867). Then, kainic acid (KA), an agonist of a subtype of ionotropic glutamate receptor, was administered intraperitoneally at a concentration of 20 mg/kg to 2-month-old *Derl1^NesCre* and *Derl1^f/f* control mice, and the seizure phenotype was observed for 1 h. The degree of seizure was scored every 5 min with a modified version of previously established criteria (Racine, 1972) (see also Methods). *Derl1^NesCre* mice had greater seizure scores than control mice (Fig. 2A,B). Pharmacologic and genetic ablation of GABAergic interneurons in the hippocampus is known to be associated with increased seizure susceptibility

and epilepsy in mouse models (Cobos et al, 2005; Zipancic et al, 2010). To investigate the involvement of GABAergic interneurons in seizure susceptibility in *Derl1*[NesCre] mice, the number of somatostatin[+] or parvalbumin[+] interneurons in the hippocampus of 2-month-old mice was examined. No significant differences were found between *Derl1*[NesCre] and *Derl1*[f/f] mice, suggesting that at least hippocampal interneurons are not substantially affected by Derlin-1 knockout (Fig. EV2A–F). In summary, although the effect of Derlin-1 deficiency on other brain regions associated with seizure susceptibility cannot be completely excluded, appropriate Derlin-1-mediated localization of newly generated neurons may be important for reducing seizure susceptibility.

## Requirement of Derlin-1 for maintenance of NSC numbers and cognitive function in the aged mouse brain

To examine whether the decrease in the return to quiescence in NSCs and increase in the persistence of NSCs in *Derl1*[NesCre] mice affect the maintenance of neurogenesis throughout life, we quantified the number of NSCs in middle-aged (9-month-old) mice. In the *Derl1*[NesCre] mice, the number of NSCs in the DG was markedly lower than that in the control mice (Fig. 2C,D). The number of DCX[+] cells was also decreased in the *Derl1*[NesCre] mice (Fig. 2E,F). On the basis of these results, Derlin-1 may be required to maintain the NSC pool in the aged mouse brain and to ensure adequate neurogenesis throughout life. Adult hippocampal neurogenesis is vital for cognitive function, and the novel location recognition test, which uses spatial discrimination ability as an index, is known to reflect hippocampus-dependent cognitive function (Goncalves et al, 2016; Goodman et al, 2010). We investigated the temporal dynamics of NSC depletion prior to 9 months of age and found that the DGs of *Derl1*[NesCre] mice were already depleted of NSCs at 4 months of age (Fig. EV2G). Therefore, we performed a novel location recognition test in mice at 4 months of age (Fig. 2G). Compared with control mice, 4-month-old *Derl1*[NesCre] mice showed a reduced preference for the displaced object (DO) in the testing phase (Fig. 2H), suggesting that they were unable to identify changes in the locations of objects. In contrast, *Derl2*[NesCre] mice with unchanged adult neurogenesis tended to spend more time with the DO in the testing phase, similar to control mice (Fig. EV2H). These results suggest that hippocampus-dependent cognitive function is impaired in *Derl1*[NesCre] mice due to disruptions in adult neurogenesis.

## Requirement of Derlin-1 for the transition of NSCs from active to quiescent states

To elucidate the mechanism by which activated NSCs are less likely to return to a quiescent state in the DG of *Derl1*[NesCre] mice (Fig. 1Q,R), we used cultured adult rat hippocampal NSCs. Adult rat hippocampal NSCs reportedly exhibit a highly proliferative state when treated with basic fibroblast growth factor (bFGF), and treatment with diazepam or BMP4 artificially induces NSCs to reach a quiescent state (Doi et al, 2021; Mira et al, 2010; Mukherjee et al, 2016). We used these culture conditions to examine the effect of Derlin-1 deficiency on the transition of NSCs from active to quiescent states in vitro. Adult rat hippocampal NSCs transfected with anti-Derl1 siRNA were cultured for 2 days in proliferation medium, for another 2 days in proliferation medium or diazepam- or BMP4-containing quiescence medium, and analyzed after 30 min of 5-ethynyl-2-deoxyuridine (EdU) treatment (Fig. 3A).

The percentage of EdU[+] proliferating NSCs among *Derl1* knockdown (siDerl1) NSCs was unchanged under proliferating conditions compared to that among control (siControl) NSCs (Fig. 3B,C). In contrast, the percentage of EdU[+] proliferating siDerl1 NSCs increased under quiescent conditions (Fig. 3B,C). A similar result was observed with adult mouse hippocampus-derived NSCs (Fig. EV3A–C). The results of these experiments suggest that the defects in the transition of NSCs from the active to the quiescent state observed in the *Derl1*[NesCre] mice can be reproduced in vitro. To investigate whether factors secreted from Derlin-1-deficient NSCs inhibit the transition from active to quiescent states, the culture medium of activated wild-type NSCs was replaced with 50% culture medium from siControl NSCs or siDerl1 NSCs and 50% new quiescent medium (Fig. EV3F). The proliferating cell ratio was unchanged (Fig. EV3G), suggesting that Derlin-1 deficiency did not result in the secretion of factors that dominantly inhibit the transition from active to quiescent states. We next examined whether factors secreted by wild-type NSCs during quiescence induction were sufficient to improve the inhibition of the transition from active to quiescent states in Derlin-1-deficient NSCs (Fig. EV3H). The percentage of proliferating NSCs in the siDerl1 NSC population was greater than that in the siControl NSC population, even when 50% of culture medium from wild-type NSCs was used (Fig. EV3I). These data suggest that Derlin-1 regulates the transition of NSCs from active to quiescent states primarily through a cell-autonomous mechanism.

## Requirement of Stat5b expression for maintenance of NSCs

To understand the mechanism by which Derlin-1 deficiency impairs the transition of NSCs from active to quiescent states, RNA sequencing (RNA-seq) was performed on adult rat hippocampal NSCs induced to enter a quiescent state (Fig. EV4A). In siDerl1 NSCs, 184 genes were significantly upregulated (>1.5-fold), and 180 genes were downregulated (<0.8-fold) relative to siControl NSCs (Appendix Table S1). Although Derlin-1 deficiency increased the expression of ER stress-related genes in the DG (Fig. EV1B), significant enrichment of ER stress-related genes among downstream targets of Derlin-1 was not observed in adult rat hippocampal NSCs (Fig. EV4B). Therefore, it is conceivable that the abnormal transition of siDerl1 NSCs from active to quiescent states may not be triggered by ER stress itself but rather by changes in unconventional genes regulated by Derlin-1. Since it is well known that many transcription factors regulate the expression of genes involved in stem cell states, the group of genes whose expression is altered by Derlin-1 deficiency was searched in the bracket of transcription factors (Appendix Table S1) (Sarkar and Hochedlinger, 2013; Takashima and Suzuki, 2013). The expression levels of 6 transcription factors were increased in siDerl1 NSCs, while those of 9 transcription factors were decreased (Fig. EV4C). Among these transcription factors, Stat5b has been reported to be involved in the maintenance of tissue stem cell quiescence (Kollmann et al, 2021; Wang et al, 2019; Wang et al, 2009). The expression of *Stat5b* was decreased in siDerl1 NSCs derived from adult rat hippocampus and adult mouse hippocampus in both proliferative and quiescent states (Figs. 4A,B and EV3D,E). Additionally, the expression of the Stat5b protein was confirmed to be lower in siDerl1 NSCs than in siControl NSCs (Fig. 4C). These

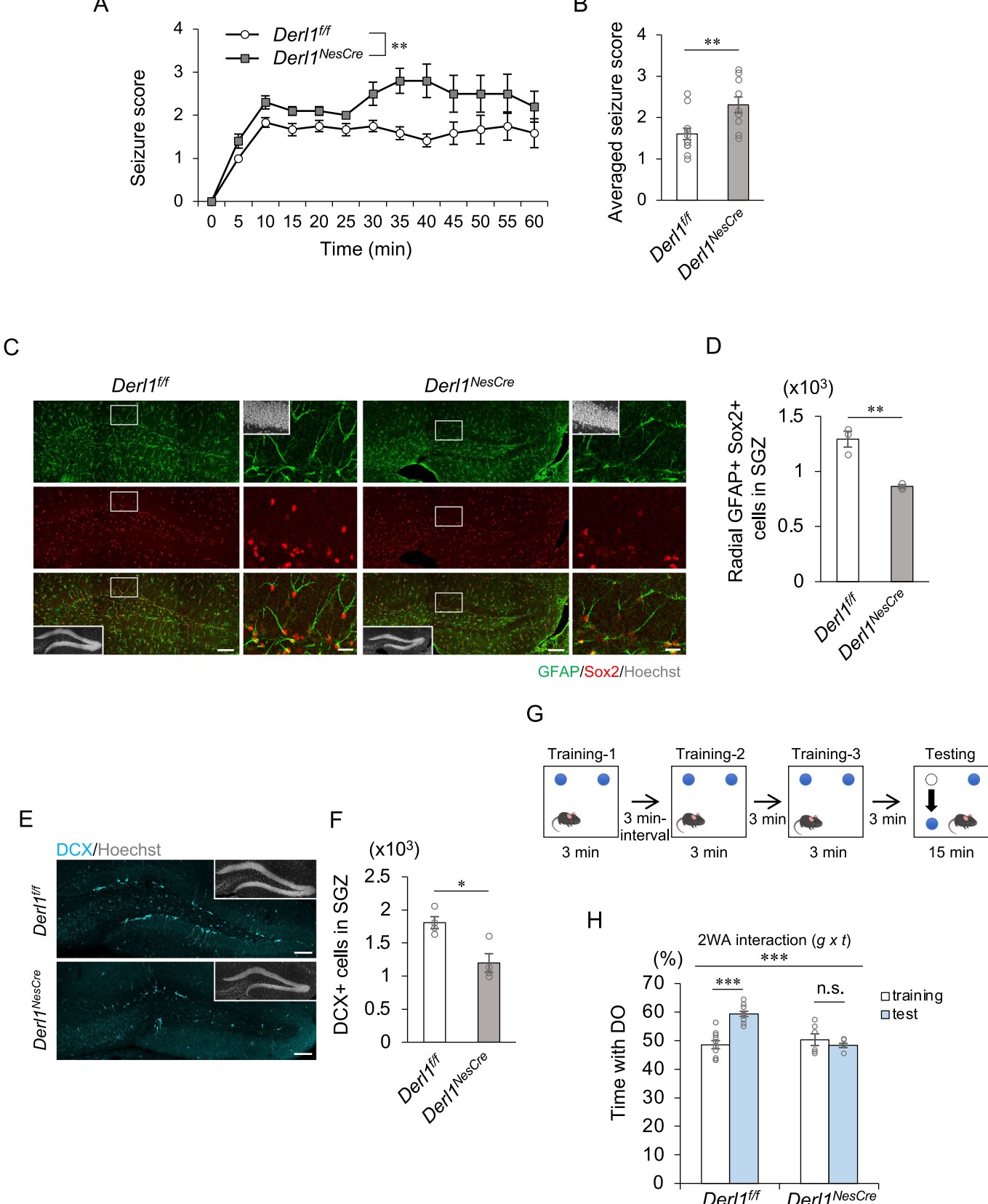

GFAP/Sox2/Hoechst

**Figure 2. Derlin-1 deficiency increases seizure susceptibility and impairs cognitive function.**

(A) Time plot showing the mean seizure score over 1 h after KA treatment in *Derl1*^f/f^ and *Derl1*^NesCre^ mice (n = 12; *Derl1*^f/f^ mice, n = 10; *Derl1*^NesCre^ mice). (B) Bar graph showing averaged seizure scores in *Derl1*^f/f^ and *Derl1*^NesCre^ mice (n = 12; *Derl1*^f/f^ mice, n = 10; *Derl1*^NesCre^ mice). (C) Representative immunofluorescence images of the DG with GFAP (green), Sox2 (red), and Hoechst staining (gray; insets). The areas outlined by a white rectangle are enlarged to the right. Scale bars, 100 μm (left images) and 20 μm (right images). (D) Quantification of the number of radial GFAP⁺ Sox2⁺ NSCs in the SGZ of 9-month-old *Derl1*^f/f^ and *Derl1*^NesCre^ mice (n = 3 mice). (E) Representative immunofluorescence images of the DG stained for DCX (cyan) in 9-month-old *Derl1*^f/f^ and *Derl1*^NesCre^ mice. Scale bars: 100 μm. (F) Quantification of the number of DCX⁺ immature neurons in the SGZ of 9-month-old *Derl1*^f/f^ and *Derl1*^NesCre^ mice (n = 4 mice). (G) Schematic diagram of the experimental protocol for the novel location recognition test. (H) Percentage of time spent with the displaced object (DO) during the training and testing phases in 4-month-old *Derl1*^f/f^ and *Derl1*^NesCre^ mice (n = 10; *Derl1*^f/f^ mice, n = 6; *Derl1*^NesCre^ mice). 2WA two-way ANOVA, g genotype, t trial. Bar graphs are presented as the mean ± SEM. *P < 0.05, **P < 0.01, and ***P < 0.001 by two-way repeated-measures ANOVA (A), Student's *t*-test (B, D, F) or two-way ANOVA followed by Tukey's test (H). n.s. not significant. Source data are available online for this figure.

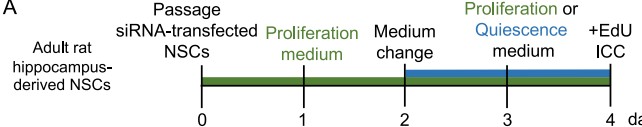

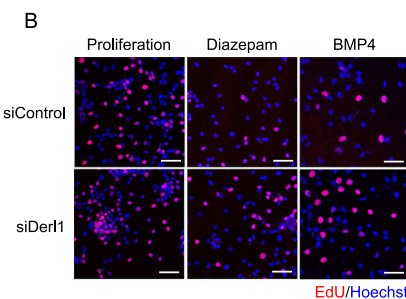

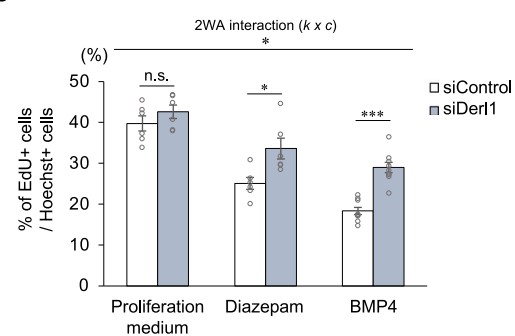

**Figure 3. Derlin-1 is required for the transition of NSCs from active to quiescent states.**

(A) Experimental scheme to induce the transition of control and *Derl1* knockdown NSCs from active to quiescent states. (B) Representative images of EdU (red) and Hoechst (blue) staining in control (siControl) and *Derl1* knockdown (siDerl1) NSCs with or without induction of quiescence with diazepam or BMP4 for 2 days. NSCs were fixed 30 min after the addition of EdU. Scale bars: 50 μm. (C) Quantification of the percentage of EdU⁺ proliferating NSCs among total Hoechst⁺ cells in siControl and siDerl1 NSCs with proliferative conditions or induction of quiescence with diazepam or BMP4 for 2 days (n = 6 biological replicates; proliferation medium and diazepam condition, n = 9 biological replicates; BMP4 condition). 2WA two-way ANOVA, k knockdown, c condition. Bar graphs are presented as the mean ± SEM. *P < 0.05 and ***P < 0.001 by two-way ANOVA followed by Tukey's test. n.s. not significant. Source data are available online for this figure.

findings suggest that the expression of Stat5b is transcriptionally regulated downstream of Derlin-1. We hypothesized that decreased expression of Stat5b may be responsible for the disruption of homeostasis in siDerl1 NSCs. To test this hypothesis, adult rat hippocampal NSCs transfected with an siRNA against *Stat5b* were cultured for 2 days in a proliferation medium and for 2 days in a quiescence medium containing BMP4 (Fig. 4A). Intriguingly, we found that Stat5b deficiency increased the percentage of EdU⁺ or Ki67⁺ proliferating adult rat hippocampal NSCs (Fig. 4D–F). This increase in the percentage of EdU⁺ proliferating cells after induction of quiescence was consistent with the results from siDerl1 NSCs (Fig. 3B,C). To examine whether the expression of Stat5b is sufficient to restore the impaired transition of siDerl1 NSCs, NSCs infected with a control lentivirus encoding Venus (a GFP variant) or a lentivirus encoding Venus-tagged Stat5b were induced to enter the quiescent state (Fig. 4G). We confirmed that all the viruses used against siControl or siDerl1 NSCs infected more than 70% of the NSCs (Fig. EV4D). We found that exogenously expressed Stat5b restored the percentage of proliferating siDerl1 NSCs to that of proliferating siControl NSCs (Fig. 4H,I), suggesting that Stat5b expressed downstream of Derlin-1 regulates the transition of NSCs from active to quiescent states.

Stat5b is a member of the Stat family of proteins that are phosphorylated by receptor-bound Janus kinase (JAK) in response to cytokines and growth factors to form homodimers or heterodimers that translocate into the nucleus to act as transcriptional activators (Able et al, 2017). To examine whether the transcriptional activity of Stat5b is required for restoring the impaired transition of siDerl1 NSCs from active to quiescent states, NSCs were infected with a virus encoding the mutant Stat5b (Y699F), in which the tyrosine phosphorylation sites required for activation were replaced with phenylalanine. We found that exogenously expressed Stat5b (Y699F) also reduced the abnormal proliferation of siDerl1 NSCs (Fig. EV4E). Moreover, downstream target genes of Stat5b were not enriched among the downstream targets of Derlin-1 in adult rat hippocampal NSCs (Fig. EV4F), suggesting that the transcriptional activity of Stat5b may not be required for NSC maintenance. We next examined whether exogenous expression of Stat5b suppresses the abnormal proliferation of Derlin-1-deficient NSCs in vivo. The DG of the hippocampus of 9-week-old *Derl1*^NesCre^ mice was infected with a lentivirus encoding control HA-tagged Venus (HA-Venus) on one side or a lentivirus encoding HA-tagged Stat5b (HA-Stat5b) on the other side by intracerebral injection, after which the mice were analyzed by immunostaining (Fig. 4J). Compared with those of the control virus-infected DG, the DG infected with the Stat5b-encoding virus had a reduced number

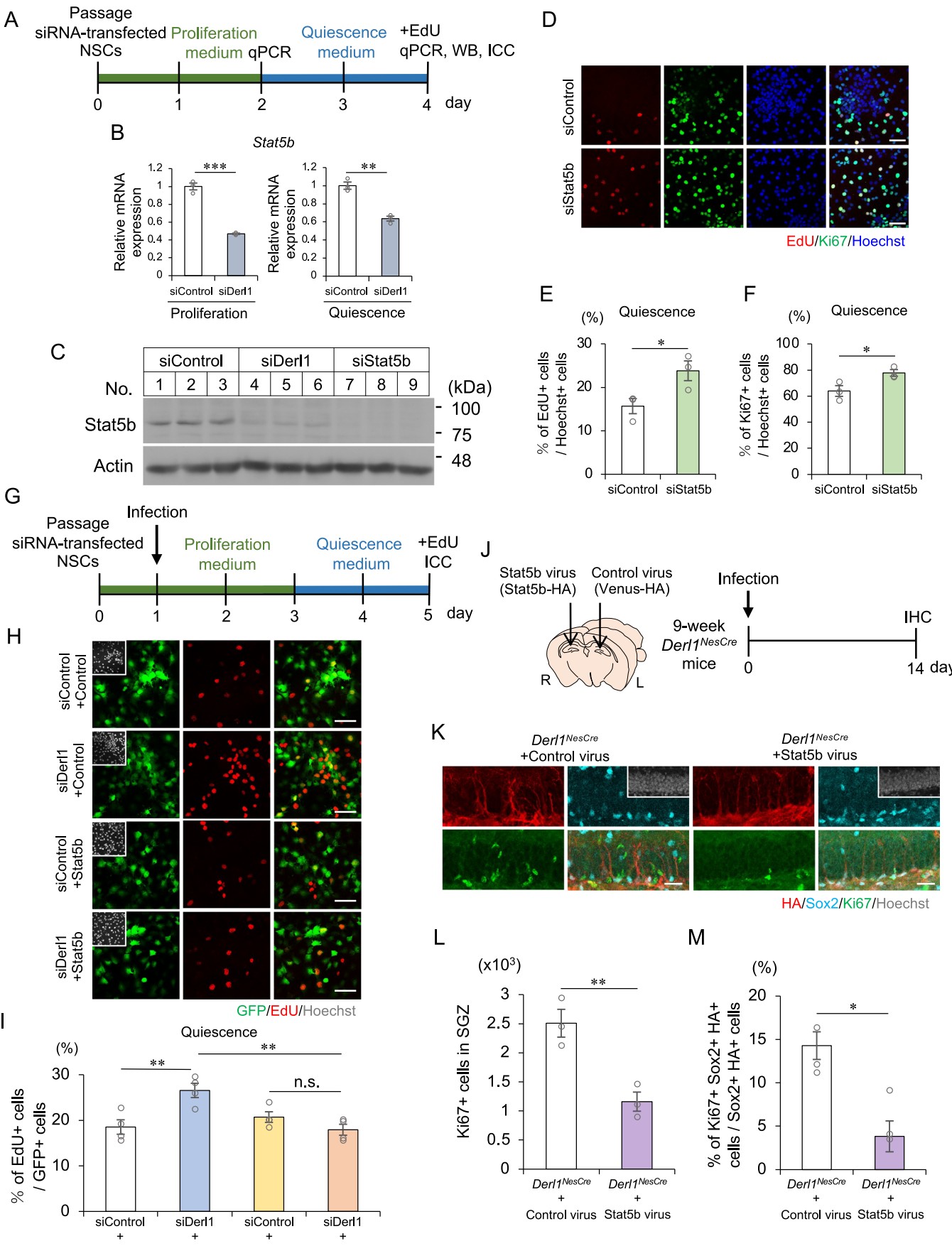

**Figure 4. Stat5b is required for the transition of NSCs from active to quiescent states.**

(A) Experimental scheme for evaluating the relevance of Stat5b underlying the impairment of NSC transition to quiescence by *Derl1* knockdown. (B) Expression of *Stat5b* in siControl and siDerl1 NSCs under proliferation and quiescent conditions. Gene expression levels were estimated by qPCR and normalized to that of *β-actin* ($n = 3$ biological replicates). (C) Representative immunoblots (IB) of siControl, siDerl1, and siStat5b NSCs after induction of quiescence. Whole-cell lysates were analyzed by IB with Stat5b and actin antibodies. (D) Representative images of EdU (red), Ki67 (green), and Hoechst (blue) staining in siControl and siStat5b NSCs after induction of quiescence with BMP4 for 2 days. NSCs were fixed 30 min after the addition of EdU. Scale bars: 50 m. (E, F) Quantification of the percentages of EdU⁺ (E) and Ki67⁺ (F) proliferating NSCs among total Hoechst⁺ cells in siControl and siStat5b NSCs after induction of quiescence with BMP4 for 2 days ($n = 3$ biological replicates). (G) Experimental scheme for investigating the requirement of Stat5b for impairment of NSC transition to quiescence by *Derl1* knockdown. (H) Representative images of GFP (green), EdU (red), and Hoechst (gray; insets) staining in siControl and siDerl1 quiescence-conditioned NSCs with or without exogenous Stat5b expression. NSCs were fixed 30 min after the addition of EdU. Scale bars: 50 μm. (I) Quantification of the percentage of EdU⁺ proliferating NSCs among total GFP⁺ cells in siControl and siDerl1 NSCs with or without exogenous Stat5b expression ($n = 4$ biological replicates). (J) Experimental scheme for assessing the effect of Stat5b expression in the DG on NSC proliferation in *Derl1^{NesCre}* mice. (K) Representative immunofluorescence images of the DG with HA (red), Sox2 (cyan), Ki67 (green), and Hoechst staining (gray; insets). Scale bars: 25 μm. (L) Quantification of the number of Ki67⁺ proliferating cells in the SGZ of *Derl1^{NesCre}* mice with or without exogenous Stat5b expression ($n = 3$ mice). (M) Quantification of the percentage of Ki67⁺ Sox2⁺ HA⁺ proliferating NS/PCs among total Sox2⁺ HA⁺ cells in the DG of *Derl1^{NesCre}* mice with or without exogenous Stat5b expression ($n = 3$ mice). Bar graphs are presented as the mean ± SEM. *$P < 0.05$, **$P < 0.01$, and ***$P < 0.001$ by Student's *t*-test (B, E, F, L, M) or one-way ANOVA followed by Bonferroni's post hoc test (I). Source data are available online for this figure.

of Ki67⁺ proliferating cells and a reduced percentage of proliferating NS/PCs among virus-infected HA-Stat5b- or HA-Venus-expressing NS/PCs (Fig. 4K–M). Taken together, these results suggest that the reduced expression of Stat5b is responsible for the abnormal proliferation of NSCs in the *Derl1^{NesCre}* mice.

## Restoration of impaired transition of Derlin-1-deficient NSCs from active to quiescent states by 4-PBA

We previously reported that continuous treatment of *Derl1^{NesCre}* mice with 4-PBA improved motor impairment due to brain atrophy (Sugiyama et al, 2021). 4-PBA acts not only as a chemical chaperone but also as an HDAC inhibitor, and we and others have shown that other HDAC inhibitors, such as valproic acid (VPA), counteract neurological diseases, including spinal cord injury and hearing loss, in mouse models by promoting neuronal differentiation (Abematsu et al, 2010; Kusaczuk et al, 2015; Wakizono et al, 2021). Furthermore, VPA is known to inhibit NS/PC proliferation by inducing the cyclin-dependent kinase inhibitor p21 through its HDAC inhibitor activity (Jessberger et al, 2007). Based on these findings, to verify the possibility that 4-PBA may be effective at rescuing the abnormality of Derlin-1-deficient NSCs, NSCs were pretreated with 4-PBA one day before induction to a quiescent state, and the percentage of proliferating NSCs was quantified 3 days later (Fig. 5A). The percentage of proliferating siDerl1 NSCs showed a trend of decrease by 4-PBA treatment (Fig. 5B,C). We examined the effect of 4-PBA treatment on *Stat5b* expression in NSCs (Fig. 5D). The expression of *Stat5b* was increased in siControl and siDerl1 NSCs by 4-PBA treatment, and the reduced *Stat5b* expression in siDerl1 NSCs recovered to the same level found in the vehicle-treated siControl NSCs (Fig. 5E). We then examined whether Stat5b expression is required for the effect of 4-PBA treatment on the impaired transition of Derlin-1-deficient NSCs from active to quiescent states. In NSCs with simultaneous knockdown of *Derl1* and *Stat5b*, the rescue effect of 4-PBA was abolished (Fig. 5F,G), suggesting that 4-PBA ameliorates the aberrant proliferation of Derlin-1-deficient NSCs via Stat5b expression.

We next examined whether this induction of *Stat5b* expression depended on chaperone activity or HDAC inhibition activity by using other chemical chaperones, tauroursodeoxycholic acid (TUDCA) and trehalose, and the representative HDAC inhibitor

VPA (Fig. EV5A). Treatment with TUDCA and trehalose did not affect *Stat5b* expression in either siControl or siDerl1 NSCs (Fig. EV5B,C). In contrast, *Stat5b* expression was significantly increased by VPA treatment but not by treatment with valpromide (VPM), a VPA analog that has no inhibitory effect on HDACs (Fig. EV5D,E). Consistent with the *Stat5b* expression results, the percentage of proliferating siDerl1 NSCs was reduced by treatment with VPA but not TUDCA (Fig. EV5F–H). These results suggest that the HDAC inhibitor activity of 4-PBA contributes to increased *Stat5b* expression and thus may rescue the impaired transition of Derlin-1-deficient NSCs from active to quiescent states.

## Amelioration of abnormal adult neurogenesis and associated pathological phenotypes by 4-PBA treatment in Derlin-1-deficient mice

4-PBA can cross the blood–brain barrier easily and has been confirmed to exert a therapeutic effect on mouse models of neurological diseases such as Alzheimer's disease and ALS (Ryu et al, 2005; Wiley et al, 2011). We reported that 4-PBA administration improved brain atrophy and motor dysfunction in *Derl1^{NesCre}* mice (Sugiyama et al, 2022). Therefore, we administered 4-PBA intraperitoneally to *Derl1^{f/f}* and *Derl1^{NesCre}* mice for 14 days and examined its effect on adult hippocampal neurogenesis in vivo (Fig. 6A). Two weeks of 4-PBA administration ameliorated the abnormal increase in the proliferation of NSCs and the ectopic localization of immature neurons in the *Derl1^{NesCre}* mice (Fig. 6B,C). To investigate whether the amelioration of abnormal neurogenesis in *Derl1^{NesCre}* mice by 4-PBA depends on HDAC inhibition, VPA was administered intraperitoneally to the *Derl1^{NesCre}* mice for 14 days (Fig. EV5I). VPA treatment mitigated the aberrant proliferation of NSCs and the ectopic localization of immature neurons in *Derl1^{NesCre}* mice, similar to the effects observed with 4-PBA treatment (Fig. EV5J–M). Based on these results, it is conceivable that the HDAC inhibitory activity of 4-PBA contributes to ameliorating the abnormal adult neurogenesis of *Derl1^{NesCre}* mice.

We then examined the effect of 4-PBA on age-related abnormalities and pathological phenotypes in *Derl1^{NesCre}* mice. When mice intraperitoneally injected with 4-PBA were subjected to the KA-induced seizure susceptibility test (Fig. 6D), 4-PBA treatment alleviated the increase in seizure score in the *Derl1^{NesCre}*

A

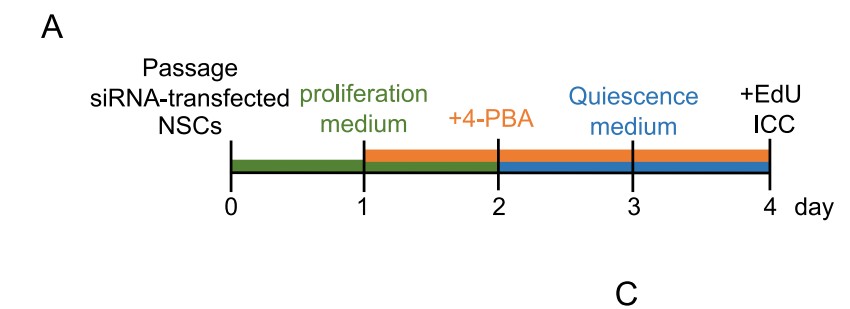

Passage
siRNA-transfected
NSCs

proliferation
medium

+4-PBA

Quiescence
medium

+EdU
ICC

0    1    2    3    4    day

B

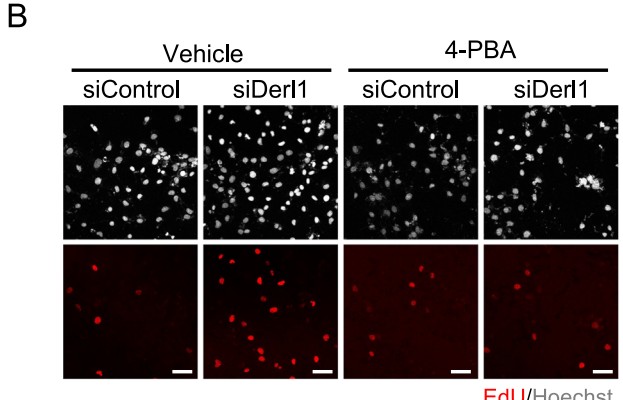

Vehicle

siControl    siDerl1

4-PBA

siControl    siDerl1

EdU/Hoechst

C

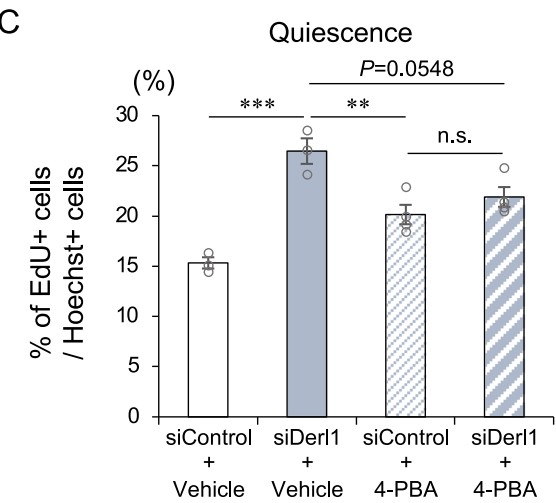

Quiescence

(%)

% of EdU+ cells / Hoechst+ cells

30
25
20
15
10
5
0

*** ** n.s. P=0.0548

siControl + Vehicle    siDerl1 + Vehicle    siControl + 4-PBA    siDerl1 + 4-PBA

D

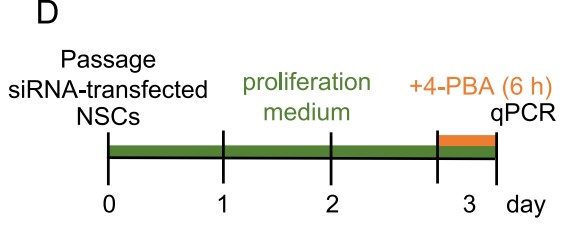

Passage
siRNA-transfected
NSCs

proliferation
medium

+4-PBA (6 h)
qPCR

0    1    2    3    day

E

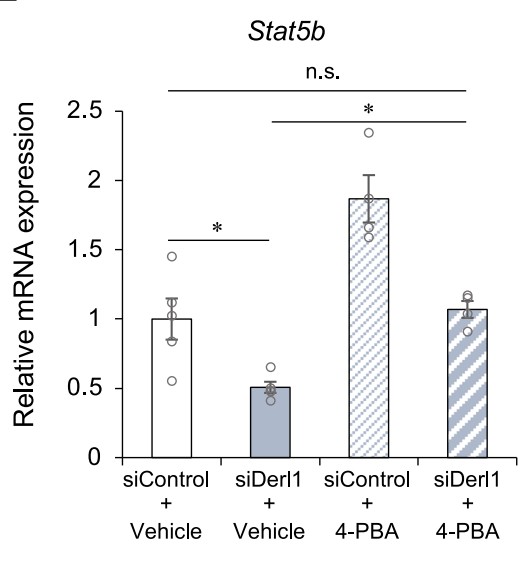

*Stat5b*

Relative mRNA expression

2.5
2
1.5
1
0.5
0

n.s.
*
*

siControl + Vehicle    siDerl1 + Vehicle    siControl + 4-PBA    siDerl1 + 4-PBA

F

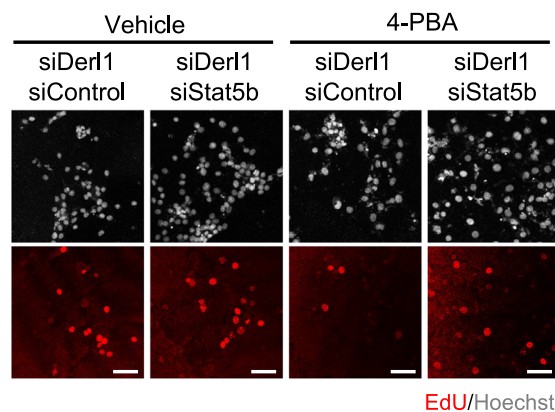

Vehicle

siDerl1 siControl    siDerl1 siStat5b

4-PBA

siDerl1 siControl    siDerl1 siStat5b

EdU/Hoechst

G

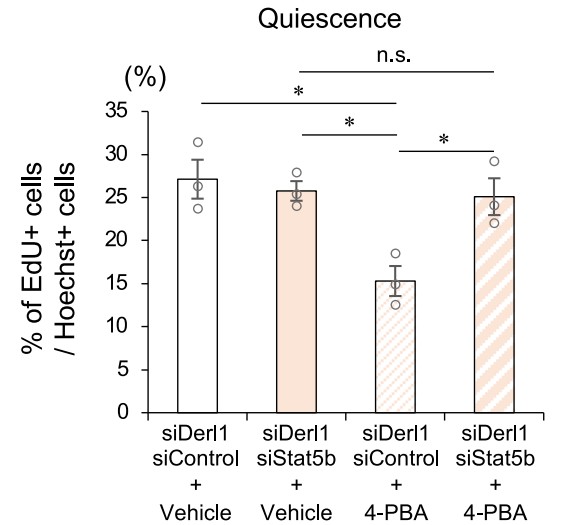

Quiescence

(%)

% of EdU+ cells / Hoechst+ cells

35
30
25
20
15
10
5
0

n.s.
*
*
*

siDerl1 siControl + Vehicle    siDerl1 siStat5b + Vehicle    siDerl1 siControl + 4-PBA    siDerl1 siStat5b + 4-PBA

Figure 5. 4-PBA induces *Stat5b* expression in NSCs and restores the impaired transition of Derlin-1-deficient NSCs from active to quiescent states.

(A) Experimental scheme for evaluating the effect of 4-PBA on the impairment of NSC transition to quiescence by *Derl1* knockdown. (B) Representative images of Hoechst (gray) and EdU (red) staining in siControl and siDerl1 NSCs treated with or without 4-PBA (1 mM). NSCs were fixed 30 min after the addition of EdU. Scale bars: 50 μm. (C) Quantification of the percentage of EdU$^+$ proliferating NSCs among total Hoechst$^+$ cells in 4-PBA-treated siControl and siDerl1 NSCs induced to enter the quiescent state by the administration of BMP4 for 2 days ($n = 3$ biological replicates; Vehicle, $n = 4$ biological replicates; 4-PBA). (D) Experimental scheme for assessing the expression of *Stat5b* in siControl and siDerl1 NSCs treated with or without 4-PBA. (E) Expression of *Stat5b* in siControl and siDerl1 NSCs with or without 4-PBA (1 mM) treatment. Gene expression levels were estimated by qPCR and normalized to that of *β-actin* ($n = 5$ biological replicates; Vehicle, $n = 4$ biological replicates; 4-PBA). (F) Representative images of Hoechst (gray) and EdU (red) staining in siDerl1, siControl, and siStat5b NSCs treated with or without 4-PBA (1 mM). NSCs were fixed 30 min after the addition of EdU. Scale bars: 50 μm. (G) Quantification of the percentage of EdU$^+$ proliferating NSCs among total Hoechst$^+$ cells in 4-PBA-treated siDerl1, siControl, and siDerl1, siStat5b NSCs induced to enter the quiescent state by the administration of BMP4 for 2 days ($n = 3$ biological replicates). Bar graphs are presented as the mean ± SEM. *$P < 0.05$, **$P < 0.01$, and ***$P < 0.001$ by one-way ANOVA followed by Bonferroni's post hoc test. n.s. not significant. Source data are available online for this figure.

mice (Fig. 6E). Next, we assessed the effect of 4-PBA on NSC depletion and cognitive dysfunction in aged *Derl1$^{NesCre}$* mice. Because long-term intraperitoneal injection stresses the mice, 4-PBA was administered to *Derl1$^{f/f}$* and *Derl1$^{NesCre}$* mice via water supplied ad libitum from 4 weeks to 16–20 weeks of age (Fig. 6F). The number of NSCs in the DG of 4-PBA-treated *Derl1$^{NesCre}$* mice recovered to a similar level as that in 4-PBA-treated *Derl1$^{f/f}$* mice, suggesting that the aging-dependent depletion of Derlin-1-deficient NSCs is ameliorated by 4-PBA treatment (Fig. 6G). To examine the effect of 4-PBA treatment on the impaired hippocampus-dependent cognitive function of *Derl1$^{NesCre}$* mice, we performed a novel location recognition test and found that the preference for the DO in the testing phase was restored in 4-PBA-treated *Derl1$^{NesCre}$* mice, as observed in *Derl1$^{f/f}$* mice (Fig. 6H). These results suggest that the impairment of adult hippocampal neurogenesis caused by Derlin-1 deficiency and associated pathological phenotypes, i.e., increased seizure susceptibility and cognitive dysfunction, can be ameliorated by the administration of 4-PBA.

## Discussion

In the present study, we show that Derlin-1 is required for the proper proliferation of NSCs and localization of newborn neurons in the DG through the expression of Stat5b and for brain functions associated with adult hippocampal neurogenesis, including seizure suppression and cognitive function. Furthermore, 4-PBA was found to be effective at rescuing the detrimental phenotypes of Derlin-1-deficient mice via HDAC inhibition.

Maintaining a delicate balance between the quiescent and active states of NSCs is crucial in the adult mammalian brain to prevent depletion and ensure a continuous generation of an adequate number of neurons throughout life (Bond et al, 2015; Encinas et al, 2011; Shin et al, 2015). Our finding that decreased Stat5b expression increases the number of activated NSCs is consistent with previous reports that Stat5b is required for the maintenance of quiescence in tissue stem cells such as hematopoietic stem cells and hair follicle stem cells (Wang et al, 2019; Wang et al, 2009). The tyrosine phosphorylation involved in the transcriptional activation of Stat5b was not required to rescue the abnormal proliferation of Derlin-1-deficient NSCs, and downstream target genes of Stat5b were not enriched among the downstream target genes of Derlin-1. Thus, it is conceivable that the transcriptional activity of Stat5b may not be required to maintain the quiescent state of NSCs. Previous studies have shown that Stat5b, in addition to its role as a

transcription factor, is localized to the ER in smooth muscle cells and human pulmonary arterial endothelial cells and is important for maintaining ER structure and mitochondrial function as a nongenomic effect (Lee et al, 2012; Lee et al, 2013; Sehgal, 2013). Additionally, Stat5-family proteins without tyrosine phosphorylation are localized in the nucleus and are known to be involved in cytokine-induced megakaryocyte differentiation (Park et al, 2016). Although it is possible that Stat5b may act as a phosphorylation-independent transcription factor or a transcriptional activity–independent factor in the maintenance of NSCs, the Derlin-1-Stat5b axis is an indispensable pathway in adult hippocampal neurogenesis. Another important issue that remains to be clarified is how Derlin-1 transcriptionally regulates Stat5b expression. Derlin-1 is an ERAD component and is indispensable for ER quality control, but has no transcriptional activity. The most likely possibility is that the UPR caused by Derlin-1 deficiency inhibits the expression of *Stat5b* mRNA in the activated NSCs. Although ER stress was also induced in the DG of Derlin-2-deficient mice (Fig. EV1N) (Sugiyama et al, 2021), abnormal proliferation of NSCs was not observed in Derlin-2-deficient mice (Fig. EV1O,P). On the other hand, no changes in ER stress-related gene expression were observed in the cultured Derlin-1-deficient NSCs (Fig. EV4B), and ER stress itself in NSCs has been reported to inhibit their proliferation (Bowers et al, 2020). Although the possible involvement of ER stress response cannot be ruled out, it is strongly suggested that Derlin-1-specific downstream targets may contribute to the transcriptional regulation of *Stat5b*.

Derlin-1 deficiency induces the ectopic localization of newborn neurons in the hilus, which may be due to abnormal migration. Among the factors involved in cell migration, the expression of CXC motif chemokine receptor 4 (Cxcr4), which is indispensable in NS/PCs, has been implicated in the appropriate localization of newborn neurons in the adult DG (Sakai et al, 2018; Schultheiss et al, 2013). Although *Cxcr4* was not found in differentially expressed genes in siDerl1 NSCs in our RNA-seq analysis (Appendix Table S1), it may be possible that the expression of Cxcr4 protein on the plasma membrane surface is suppressed by Derlin-1 deficiency. It is also possible that abnormally proliferated NSCs or newborn neurons might be physically extruded from the DG to the hilus or that Stat5b directly or indirectly regulates the location of adult neurogenesis. The mechanism by which the survival of NS/PCs, which proliferate in the adult DG, is ultimately reduced in Derlin-1-deficient mice is also unknown. While cultured Derlin-1-deficient NSCs did not exhibit heightened expression of the ER stress response gene cluster (Fig. EV4B), the expression of

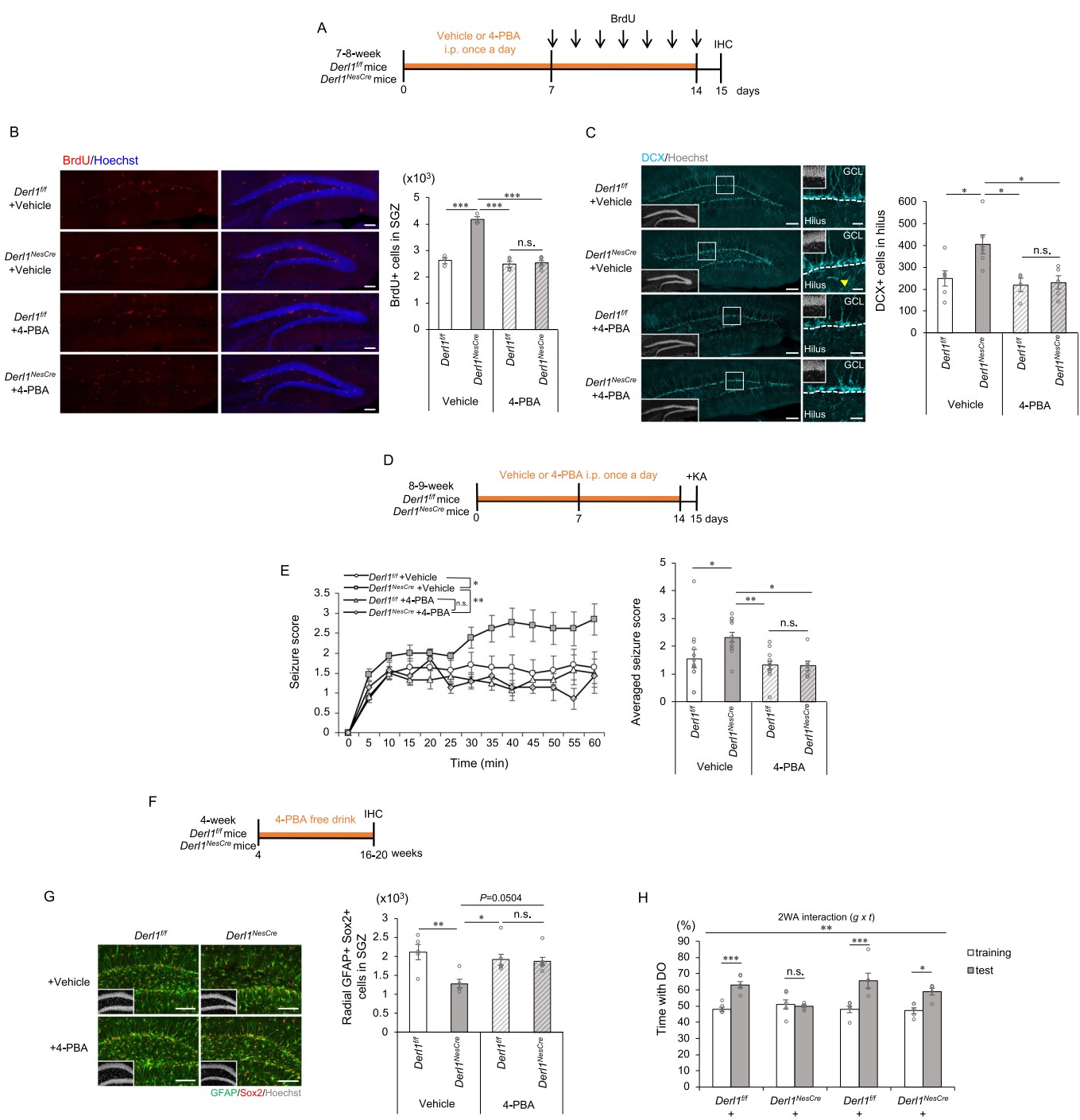

ER stress response genes, encompassing the ER stress-induced cell death-associated gene *Chop*, was heightened within the DG region of *Derl1^NesCre^* mice (Figs. EV1B, EV5N). Derlin-1-mediated maintenance of ER homeostasis may be important for the viability of newborn neurons derived from adult NSCs in the DG of the hippocampus. Although further research is needed to elucidate these unresolved mechanisms, our results suggest that Derlin-1 regulates adult neurogenesis in a spatiotemporal manner.

ER stress is thought to be involved in the pathogenesis of neurological diseases, including ALS and spinocerebellar ataxia (Ghemrawi and Khair, 2020; Nishitoh et al, 2008; Nishitoh et al, 2002). Chemical chaperone therapy is currently being developed as a treatment for several diseases, including some neurological diseases, with the aim of reducing ER stress. For example, clinical trials are currently being conducted on ALS patients using sodium phenylbutyrate (a salt of 4-PBA) and TUDCA, which have shown

**Figure 6.  4-PBA improves the increased seizure susceptibility and cognitive dysfunction in *Derl1^NesCre^* mice.**

(A) Experimental scheme for investigating the proliferation of NS/PCs in *Derl1^f/f^* and *Derl1^NesCre^* mice with or without 4-PBA treatment. *Derl1^f/f^* and *Derl1^NesCre^* mice treated with vehicle or 4-PBA daily for 2 weeks were simultaneously injected with BrdU daily for 7 days during the latter and fixed 1 day after the last BrdU injection. (B) Representative immunofluorescence images of the DG with BrdU (red) and Hoechst (blue) staining and quantification of the number of BrdU^+^ proliferating cells (right) in the SGZ of *Derl1^f/f^* and *Derl1^NesCre^* mice treated with or without 4-PBA (n = 3; + Vehicle mice, n = 4; *Derl1^f/f^* + 4-PBA mice, n = 6; *Derl1^NesCre^* + 4-PBA mice). Scale bars: 100 μm. (C) Representative immunofluorescence images of the DG with DCX (cyan) and Hoechst staining (gray; insets). The areas outlined by a white rectangle are enlarged to the right. The yellow arrowhead indicates DCX^+^ ectopic immature neurons in the hilus and dashed white lines indicate the boundaries between the GCL and hilus. Scale bars, 100 μm (left images) and 20 μm (right images) (left). Quantification of the number of DCX^+^ cells in the hilus in *Derl1^f/f^* and *Derl1^NesCre^* mice treated with or without 4-PBA (right) (n = 6; *Derl1^f/f^* + Vehicle mice and *Derl1^NesCre^* + Vehicle mice, n = 4; *Derl1^f/f^* + 4-PBA mice, n = 5; *Derl1^NesCre^* + 4-PBA mice). (D) Experimental scheme for investigating seizure susceptibility in *Derl1^f/f^* and *Derl1^NesCre^* mice treated with or without 4-PBA. (E) Time plot showing the mean seizure score over 1 h after KA treatment (left) and a bar graph showing the averaged seizure score (right) in *Derl1^f/f^* and *Derl1^NesCre^* mice treated with or without 4-PBA (n = 14; *Derl1^f/f^* + Vehicle mice, n = 13; *Derl1^NesCre^* + Vehicle mice, n = 12; *Derl1 ^f/f^* + 4-PBA mice, and n = 7; *Derl1^NesCre^* + 4-PBA mice). (F) Experimental scheme for assessing the effect of 4-PBA on the depletion of NSCs and cognitive function in *Derl1^f/f^* and *Derl1^NesCre^* mice. 4-PBA solutions were administered from 4 weeks to 16–20 weeks (4 months) of age through the water supply, which was available ad libitum. (G) Representative immunofluorescence images of the DG with GFAP (green), Sox2 (red), and Hoechst staining (gray; insets) and quantification of the number of radial GFAP^+^ Sox2^+^ NSCs in the SGZ of 4-month-old *Derl1^f/f^* and *Derl1^NesCre^* mice treated with or without 4-PBA (n = 5; + Vehicle mice, n = 7; + 4-PBA mice). Scale bars: 100 μm. (H) Percentage of time spent with the displaced object (DO) during the training and testing phases in 4-month-old *Derl1^f/f^* and *Derl1^NesCre^* mice treated with or without 4-PBA (n = 3; + Vehicle mice, n = 5; + 4-PBA mice). 2WA two-way ANOVA, g genotype, t trial. Bar graphs are presented as the mean ± SEM. *P < 0.05, **P < 0.01, and ***P < 0.001 by one-way ANOVA followed by Bonferroni's post hoc test [B, C, E, G (right)], two-way repeated-measures ANOVA [E (left)] or two-way ANOVA followed by Tukey's test (H). n.s not significant. Source data are available online for this figure.

effects such as delayed disease progression and prolonged survival (Paganoni et al, 2022; Paganoni et al, 2020). However, it is questionable whether 4-PBA improves the pathology of neurological disease through its chaperone activity alone. Other chemical chaperones, TUDCA, and trehalose, had no effect on *Stat5b* expression, while VPA increased its expression, suggesting that HDAC inhibition by 4-PBA may function to restore *Stat5b* expression in Derlin-1-deficient NSCs. Since both 4-PBA and VPA are short-chain fatty acid group HDAC inhibitors that mainly inhibit class I HDACs (HDAC1, 2, 3, and 8) (de Ruijter et al, 2003), it is possible that activation of class I HDACs in NSCs may suppress *Stat5b* expression. In this study, we discovered a novel function of the HDAC inhibitor 4-PBA in regulating adult neurogenesis by inducing specific genes, including *Stat5b*. Although further studies are needed to elucidate the precise molecular mechanisms by which HDAC inhibitors ameliorate abnormal adult neurogenesis in Derlin-1-deficient mice, this study demonstrates that the administration of HDAC inhibitors such as 4-PBA and VPA may be applicable in research aiming to clarify the pathological mechanisms of diseases caused by the disruption of adult neurogenesis.

In summary, the Derlin-1-Stat5b axis is essential for maintaining adult hippocampal neurogenesis throughout life. Maintenance of adult hippocampal neurogenesis via Derlin-1 function is essential for controlling seizure susceptibility and maintaining cognitive function, and pathologies caused by its disruption may be ameliorated by HDAC inhibition. Our discovery paves the way for the elucidation of mechanisms and the possible treatment of neurological diseases caused by abnormal adult neurogenesis.

## Methods

### Animals

All mice used in this experiment were raised under specific-pathogen-free conditions, housed under a 12-h/12-h light/dark cycle, and fed ad libitum. Details regarding *Derl1^f/f^* mice, *Derl2^f/f^* mice, and mice expressing Cre recombinase driven by the *nestin* or

*CaMKIIa* promoter have been described in previous reports (Dougan et al, 2011; Isaka et al, 1999; Karpati et al, 2019; Sugiyama et al, 2021). These mice were intercrossed to generate *Derl1^NesCre^* mice, *Derl1^CaMKIIαCre^* mice, and *Derl2^NesCre^* mice. Both male and female mice were used. All mouse experiments were approved by the Animal Research Committee of the University of Miyazaki following institutional guidelines. The experiments were conducted according to institutional guidelines. All efforts were made to minimize animal suffering and reduce the number of animals used.

### Cell lines

Human embryonic kidney (HEK) 293T cells were obtained from the American Type Culture Collection (ATCC). HEK293T cells were grown in Dulbecco's modified Eagle's medium (DMEM) (08459-64, Nacalai Tesque) supplemented with 10% FBS and penicillin–streptomycin solution (09367-34, Nacalai Tesque). Adult rat hippocampal NSCs were isolated and cloned from Fisher 344 rats and characterized in previous reports (Mira et al, 2010; Palmer et al, 1997). Adult rat hippocampal NSCs were cultured in DMEM/F12 supplemented with N2, penicillin–streptomycin solution, and bFGF (20 ng/mL) (100-18B, PeproTech) (proliferation medium) or bFGF (10 ng/mL) and diazepam (100 μM) (045-18901, Wako) or BMP4 (50 ng/mL) (5020-BP, R&D Systems) (quiescence medium) on coated culture dishes with poly-L-ornithine (P-3655, Sigma–Aldrich) and laminin (354232, Corning). Adult mouse hippocampal NSCs with multipotency and self-renewal capacity and passaged more than ten times were used in this study. The cell preparation process is as follows. Three 8-week-old C57BL/6 mice were euthanized, and each brain was promptly harvested. The DG was quickly microdissected under a dissection scope and minced with a scalpel. The dissociated DG was transferred into a prewarmed papain (25 U/mL) solution and incubated at 37 °C for 30 min. The suspension was then washed with 2 mL Minimum Essential Medium α (12571063, Gibco) containing 5% BSA to stop enzyme activity, mechanically dissociated by gentle pipetting with a fire-polished Pasteur pipette, and centrifuged at 130 × g for 5 min. The cell pellet was then resuspended in 2 mL of Hanks' balanced salt solution (HBSS) containing 250 U/mL DNase and centrifuged at

$130 \times g$ for 5 min. After centrifugation, the cells were suspended in HBSS. These cells were plated on poly-L-ornithine/laminin-coated dishes in DMEM/F12 containing N2 supplement (17502048, Gibco) with 20 ng/mL bFGF (100-18B, PeproTech), 20 ng/mL epidermal growth factor (EGF; AF-100-15, PeproTech), 1/1000x dilution of B27 (17504-044, Gibco), and 0.1 mg/mL penicillin/streptomycin/fungizone (SV30079.01, HyClone) (proliferation medium). Adult mouse hippocampal NSCs were induced to quiescence by treatment with diazepam (100 μM) (045-18901, Wako) or BMP4 (50 ng/mL) (5020-BP, R&D Systems) in a proliferation medium. All cells were maintained under a 5% $CO_2$ atmosphere at 37 °C.

## siRNA transfection

siRNA transfection was performed using Lipofectamine RNAiMAX reagent (56532, Invitrogen). The following siRNAs were used for the knockdown of adult rat- and mouse-derived hippocampal NSCs: Stealth RNAi™ siRNA Derl1-MSS289837 (Invitrogen), Stealth RNAi™ siRNA Stat5b-RSS332572 (Invitrogen). Stealth RNAi™ siRNA Negative Control Med GC Duplex (452001, Invitrogen) was used as the control. The siRNAs sequences are shown in Appendix Table S2.

## BrdU administration

To label proliferating cells, BrdU (B5002, Sigma−Aldrich) dissolved in saline (0.9% NaCl) was injected (50 mg/kg) intraperitoneally into 8- or 9-week-old mice once a day for 7 days. The mice were sacrificed 1 day, 3 weeks, or 4 weeks after the last BrdU injection. P14 mice were given 100 mg/kg BrdU intraperitoneally and sacrificed 30 min later.

## Tissue preparation for immunofluorescence

Mice were deeply anesthetized by intraperitoneal injection of a 4 mg/kg midazolam/0.3 mg/kg medetomidine/5 mg/kg butorphanol mixture and transcardially perfused with phosphate-buffered saline (PBS) followed by 4% paraformaldehyde (PFA) in PBS. Brains were dissected and postfixed overnight in 4% PFA at 4 °C. Fixed brains were incubated in 15% sucrose solution at 4 °C overnight, followed by 30% solution at 4 °C overnight. Brains were then cut into two pieces along the midline, and each half was embedded in an optimal cutting temperature compound (4583, Tissue Tek; Sakura Finetek) and stored at −80 °C. Embedded frozen brains were serially sectioned in the coronal plane at 40-μm thickness using a freezing microtome (CM3050S, Leica Microsystems). Every sixth section was sequentially transferred to 6-well plates of PBS for subsequent immunohistochemical staining.

## Immunohistochemistry

The brain sections were washed with PBS and incubated in blocking buffer (PBS containing 3% FBS and 0.1% Triton X-100) for 1 h at room temperature (RT), followed by overnight incubation at 4 °C with the primary antibody diluted in blocking buffer. Sections were washed three times with PBS and incubated for 2 h at RT with a secondary antibody diluted in a blocking buffer. After a third wash with PBS, the sections were mounted on glass slides with Immu-Mount (9990402, Thermo Scientific). For the staining of BrdU, sections were incubated with 2 N HCl at 37 °C for 15 min and washed with PBS three times before being blocked. For the staining of Ki67, antigen retrieval was performed by heating sections in target retrieval solution (S1699, DAKO) at 105 °C for 15 min before blocking. Immunofluorescence images were acquired using a confocal laser microscope (TSC-SP8, Leica Microsystems) and processed using Adobe Photoshop Elements 2021 (Adobe). Nuclei were counterstained using bisbenzimide H33258 fluorochrome trihydrochloride solution (Hoechst; 19173-41, Nacalai Tesque). Antibodies are listed in Appendix Table S3.

## Cell counting in brain sections

Quantifying the number of respective marker-positive cells in the DG, SGZ, GCL, or hilus was performed using every sixth hemisphere section. A total of 10–12 brain slices collected every six slices were measured; the number of cells in all slices was summed and then multiplied by 6 to calculate the number of cells around the DG, SGZ, and hilus within a single hippocampus. The number of cells was measured in a single plane using confocal microscopy after brain slices were prepared. A cell was determined to be located in the hilus if the soma of the cell was clearly located on the hilus side relative to the continuous line drawn between the SGZ and the hilus. The numbers of marker-positive glial cells were manually counted within 10–11 $100 \times 100$ μm areas of the ML and the hilus. These cell numbers are reported per $mm^2$. Previous studies have shown that the number of NSCs per hippocampus varies from paper to paper (Bowers et al, 2020; Encinas et al, 2011; Harris et al, 2021; Jessberger et al, 2005). This may depend on the method to quantify the number of cells (e.g., whether images are taken in a Z-stack to calculate the number per slice, or whether the number is measured per DG on one side or both sides, etc.). Although the number of NSCs is relatively small compared to other groups, consistent with some previous reports (Bowers et al, 2020; Jessberger et al, 2005), the same measurement method was used throughout the manuscript.

## In vitro cell proliferation assay

To label proliferating NSCs, 10 mg/mL EdU from a Click-iT EdU Alexa Fluor 555 Imaging Kit (C10338, Invitrogen) was added to the culture medium 30 min before fixation. EdU staining was performed following the kit manufacturer's instructions, followed by immunocytochemistry (below).

## Immunocytochemistry

Adult hippocampal NSCs were fixed with 4% PFA in PBS for 20 min, washed three times in PBS after EdU staining, permeabilized, blocked with blocking buffer (PBS containing 3% FBS and 0.1% Triton X-100) for 30 min at RT, and incubated for 1.5 h at RT with the indicated primary antibody diluted in blocking buffer. Cells were washed three times with PBS and incubated for 1.5 h at RT with the secondary antibody diluted in a blocking buffer. After a third wash with PBS, cells were mounted with Immu-Mount (Thermo Scientific) on glass slides. Nuclei were counterstained using Hoechst (1:500; Nacalai Tesque). Immunofluorescence images were obtained using a confocal laser microscope (Leica

Microsystems) and processed using Adobe Photoshop Elements 2021. The antibodies are listed in the Appendix Table S3.

## Immunoblotting

The adult hippocampal NSCs were lysed in lysis buffer (20 mM Tris-HCl pH 7.5, 150 mM NaCl, 5 mM EGTA, and 1% Triton X-100) supplemented with 5 μg/mL leupeptin (43449-62, Nacalai Tesque). These whole-cell lysates were resolved by sodium dodecyl sulfate–polyacrylamide gel electrophoresis (SDS–PAGE) and blotted onto polyvinylidene fluoride (PVDF) membranes. After blocking with 5% skim milk in TBS-T (50 mM Tris-HCl pH 8.0, 150 mM NaCl, and 0.05% Tween-20), the membranes were probed with the indicated antibodies, and immunolabeling was detected using an enhanced chemiluminescence (ECL) system. The antibodies are listed in Appendix Table S3.

## Lentivirus production

Lentiviruses were produced by co-transfecting HEK293T cells with the lentivirus constructs pRRL-Venus-HA, pRRL-Stat5b-Venus, pRRL-Stat5b-HA, or pRRL-Stat5b (Y699F)-Venus, and lentivirus packaging vector constructs pMD2.G (12259, Addgene) and psPAX2 (12260, Addgene) using Polyethylenimine (PEI)-Max (24765-1, Polysciences). The culture medium was changed at 16–24 h after transfection. The supernatants were collected at 24 and 48 h after a medium change and centrifuged at 6000×$g$ overnight at 4 °C. After discarding the supernatant, the virus solution was resuspended in 1 mL of new medium per 10 cm dish (in vitro conditions), and then the virus solution was concentrated using the Lenti-X Concentrator (631231, Clontech) and suspended in PBS (in vivo conditions).

## In vitro lentiviral infection

The virus solutions were introduced into adult rat hippocampal NSCs by adding these supernatants to the culture 24 h after passaging. At 48 h after infection, the medium was replaced with a quiescence medium. The cells were cultured for another 48 h and then fixed for an EdU-labeled cell proliferation assay.

## In vivo lentiviral infection

Nine-week-old *Derl1*[NesCre] mice were anesthetized by intraperitoneal injection of a 4 mg/kg midazolam/0.3 mg/kg medetomidine/5 mg/kg butorphanol mixture. The virus suspension was injected stereotaxically into the bilateral DG using the following coordinates relative to bregma: caudal, −2.0 mm; lateral, ±1.5 mm; ventral, −2.3 mm. In each DG, 1.5 μL of lentivirus was injected over 1 min using a 5 μL Hamilton syringe. Two weeks after the lentiviral injection, the brains were fixed for immunohistochemistry. Mice lacking HA-tag-positive cells in the DG were excluded from the study.

## Quantitative real-time PCR analysis

Total RNA was isolated from the DG or adult hippocampal NSCs using RNAiso Plus (9109, Takara Bio) and reverse transcribed using RevaTra Ace qPCR RT Master Mix with gDNA Remover (FSQ-301, TOYOBO) according to the manufacturer's instructions. Quantitative PCR was performed using SYBR Green PCR Master Mix (KK4602, Kapa Biosystems) and a StepOnePlus Real-Time PCR System (Applied Biosystems). Expression levels were normalized to the expression of *S18* or *β-actin* mRNA and calculated relative to the control. The primer sequences are shown in Appendix Table S2.

## DNA microarray analysis

Total RNA was extracted from the DG using a NucleoSpin RNA kit (740955, Takara Bio) following the manufacturer's instructions. A total of 150 ng of total RNA from each sample was amplified and labeled with Cy3. Next, 600 ng Cy3-labeled cRNA was fragmented, hybridized onto the SurePrint G3 Mouse GE Ver2 platform (G4852B, Agilent Technologies), and then incubated at 65 °C while being rotated for 17 h. Data were analyzed using GeneSpring software version 14.9 (Agilent Technologies) as previously described (Komatsu et al, 2013). In brief, the microarray data were normalized by quantile normalization, and the baseline signal values were transformed to the median in all samples. Then, quality control and filtering steps were performed based on flags and expression levels. Mean signal intensities were measured in duplicate and averaged to identify genes differentially expressed among mouse lines. Data from this microarray analysis have been submitted to the NCBI Gene Expression Omnibus archive as series GSE229342. GSEA was performed using GSEA v4.1.0 (https://www.gsea-msigdb.org/gsea/index.jsp). The enrichment plot shows the distribution of genes in each set that are positively (red) and negatively (blue) correlated with Derlin-1 deficiency. The Gene Ontology (GO) terms for GSEA were obtained from the Mouse Genome Informatics (MGI) GO project (http://www.informatics.jax.org/), which provides functional annotations for mouse gene products using Gene Ontology (http://www.informatics.jax.org/vocab/gene_ontology).

## RNA-seq

RNA library construction and RNA-seq were performed using an Illumina sequencing platform (GENEWIZ). *Derl1* or its control knockdown adult rat-derived hippocampal NSCs were cultured in the presence of BMP4 for 2 days. Then, three samples of total RNA from the two cell groups were extracted for transcriptome sequencing and RNA-seq analysis. The cDNA libraries were used to construct the transcriptome sequence library in GENEWIZ (S. Plainfield, NJ) company using Illumina HiSeq X. The files containing the results were processed with a standard pipeline that included end trimming with trimommatic (Bolger et al, 2014). Then, the sequence reads were mapped to the rat reference genome (rn6) using STAR (Dobin et al, 2013). The mapped sequences were converted to expression levels (transcripts per million, TPM) and quantified using RSEM (Li and Dewey, 2011). Differential gene expression analysis was performed using edgeR (Robinson et al, 2010). A fold change <0.8 was considered downregulation, and a fold change >1.5 was considered upregulation. Data from this RNA-seq analysis have been submitted to the NCBI Gene Expression Omnibus archive as series GSE229251. GSEA was performed using GSEA v4.1.0. The enrichment plot shows the distribution of genes in each set that are positively (red) and negatively (blue) correlated with *Derl1* knockdown. The GO terms for GSEA were obtained from the Rat Genome Database (RGD) (https://rgd.mcw.edu/), which provides functional annotations for rat gene products using Gene Ontology (https://rgd.mcw.edu/GO/).

Stat5b target genes were obtained by searching ChIP-Atlas (https://chip-atlas.org), and the binding criterion was set to a maximum distance of 1 kb in either direction from the transcription start site of the target. After ChIP-Atlas scoring, the top 80 potential targets were used as the gene set for GSEA.

## Seizure behavioral assays

The occurrence of spontaneous seizures was monitored in 2-month-old *Derl1*$^{f/f}$ and *Derl1*$^{NesCre}$ mice by continuous videotaping (B09JWMC5XN, Generic) for 21 h without stimulation and under free activity. The video data from this spontaneous seizure monitoring have been submitted to Zenodo (https://zenodo.org/records/10548867). Seizures were induced in 8- to 12-week-old *Derl1*$^{f/f}$ and *Derl1*$^{NesCre}$ mice by intraperitoneal injections of 20 mg/kg KA (BML-EA123, Enzo Life Sciences) dissolved in distilled $H_2O$. The behavior of the mice was observed for 1 h after the injection, and a seizure score was recorded manually every 5 min. The seizure score was modified into five stages from the previously described criteria (Racine, 1972). Briefly, the following seizure scale was used: normal exploratory activity (0), staring and reduced locomotion (1), immobility with fast breathing/scratching behavior (2), repetitive head and limb movements (3), sustained rearing with forelimb clonus (4), and full body extension (full tonic extension) and death (5).

## Novel location recognition test

*Derl1*$^{f/f}$ and *Derl1*$^{NesCre}$ mice were placed in a white plastic chamber (45 × 45 × 43 [H] cm) that contained two identical objects in adjacent corners; the mice were allowed to explore the objects freely for 3 min and then taken back to their home cage for 3 min, completing one training session. After three repetitions of the training session, one of the objects was moved to the opposite side of the corner of the chamber and allowed to freely explore the familiar and displaced objects for 15 min (Testing session). All sessions were recorded with an overhead video, and exploration behavior was defined as activities such as sniffing and rearing against the object. The time spent exploring each object during the training and test sessions was scored manually. The exploration ratio for objects in novel locations (displaced objects) was calculated using the formula t $_{displaced}$ /(t $_{displaced}$ + t $_{familiar}$), as described previously (Mumby et al, 2002). The chamber and objects were cleaned with 70% ethanol before the next mouse was tested.

## 4-PBA administration

Intraperitoneal injections of 200 mg/kg 4-PBA (820986, MERCK or P21005, Sigma–Aldrich) were performed daily from 8–9 to 10–11 weeks of age for immunohistochemistry and seizure behavioral assays. To examine the depletion of NSCs and cognitive function, 10 mg/mL of 4-PBA (Sigma–Aldrich) solution was administered in the ad libitum water supply from 4 weeks to 16–20 weeks of age.

## VPA administration

Intraperitoneal injections of 300 mg/kg VPA (P4543, Sigma–Aldrich) were performed daily for 2 weeks from 8 to 10 weeks of age for immunohistochemistry.

## Statistical analysis

All data are presented as the means ± standard errors. Student's *t*-test was performed to compare two group means. One-way ANOVA followed by post hoc tests compared three or more group means. In two-way ANOVA, when a significant interaction was obtained, post hoc analysis was conducted. Data from the 1 h trial of seizure behavior were analyzed by two-way repeated-measures ANOVA, and post hoc analysis was performed using Bonferroni's multiple comparison test. Statistical analyses were performed using EZR software version 1.30 (Kanda, 2013) or GraphPad Prism 9 (GraphPad Software). A $P < 0.05$ (two-tailed) was considered significant for all tests.

# Data availability

DNA microarray and RNA-seq data generated in this study are deposited with the NCBI Gene Expression Omnibus archive as series GSE229342 (https://www.ncbi.nlm.nih.gov/geo/query/acc.cgi?acc=GSE229342) and GSE229251 (https://www.ncbi.nlm.nih.gov/geo/query/acc.cgi?acc=GSE229251). Video data of a stimulus-free, freely active state in *Derl1*$^{NesCre}$ mice are available on the Zenodo repository (https://zenodo.org/records/10548867). They are publicly available as of the date of publication. This paper does not report the original code. Any additional information required to reanalyze the data reported in this paper is available from the corresponding author upon request.

The source data of this paper are collected in the following database record: biostudies:S-SCDT-10_1038-S44319-024-00205-7.

# Peer review information

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

## Acknowledgements

We thank Dr. R. Kageyama (RIKEN) for *Tg(Nes-Cre)1Kag* mice, Dr. Ploegh HL (Boston Children's Hospital and Harvard Medical School) for *Derl2f/f* mice, Drs. A. Futatsugi (Kobe City College of Nursing) and K Mikoshiba (RIKEN, Shanghai Tech University and Toho University) for *TgN(a-CaMKII-nlCre)/10* mice, Dr. D. Trono (Ecole Polytechnique Fédérale de Lausanne) for lentivirus packaging vector constructs, Dr. K. Takao (University of Toyama) for technical assistance and discussion, Mr. M. Iwamoto (Kyushu University) for preparation of mouse hippocampal NSCs, and Dr. M. Nakai (University of Miyazaki) for discussion of the statistical analysis. This study was supported by the Frontier Science Research Center, University of Miyazaki. This study was supported by MEXT/JSPS KAKENHI (grant number, 22K06254 [NM], 21K06175 [HK], 23H00391 [KN], 18H02973 and 22H02954 [HN]), AMED (grant number JP22gm6410024 [HK] and JP20gm1310008 [KN]), Terumo Life Science Foundation (HN), Naito Foundation (HN), Ono Medical Research Foundation (NM), Takeda Science Foundation (HN), TMDU Nanken-Kyoten Grant Number 2022-39 (HN), and Joint Usage and Joint

Research Programs, Institute of Advanced Medical Sciences, Tokushima University (NM and HN).

## Author contributions

**Naoya Murao**: Conceptualization; Funding acquisition; Investigation; Writing—original draft. **Taito Matsuda**: Investigation; Writing—original draft. **Hisae Kadowaki**: Conceptualization; Supervision; Funding acquisition; Writing—review and editing. **Yosuke Matsushita**: Investigation; Methodology. **Kousuke Tanimoto**: Investigation; Methodology. **Toyomasa Katagiri**: Resources; Methodology. **Kinichi Nakashima**: Resources; Supervision; Writing—original draft; Writing—review and editing. **Hideki Nishitoh**: Conceptualization; Data curation; Supervision; Funding acquisition; Writing—original draft; Project administration; Writing—review and editing.

In addition to the CRediT author contributions listed above, the contributions in detail are:

Naoya Murao: conceptualization, investigation, funding acquisition, and writing—original draft. Taito Matsuda: investigation and writing—original draft. Hisae Kadowaki: conceptualization, funding acquisition, supervision, and writing—review and editing. Yosuke Matsushita and Kousuke Tanimoto: investigation and methodology. Toyomasa Katagiri: resources and methodology. Kinichi Nakashima: resources, supervision and writing—original draft, review, and editing. Hideki Nishitoh: conceptualization, data curation, supervision, funding acquisition, project administration and writing—original draft, review, and editing.

Source data underlying figure panels in this paper may have individual authorship assigned. Where available, figure panel/source data authorship is listed in the following database record: biostudies:S-SCDT-10_1038-S44319-024-00205-7.

## Disclosure and competing interests statement

The authors declare no competing interests.

# Expanded View Figures

**Figure EV1.   Loss of *Derl1*, but not *Derl2*, specifically promotes NSC activation in the adult DG.**

(A) Representative immunofluorescence images with Hoechst (gray) and Derlin-1 staining (red) in the adult hippocampus of *Derl1*^{f/f} and *Derl1*^{NesCre} mice. Scale bars: 100 µm. (B) GSEA showing differential expression of 238 genes in the DG categorized by the GO term "Response to ER stress." GSEA shows gene expression changes in the DG of *Derl1*^{NesCre} mice relative to *Derl1*^{f/f} mice. The enrichment plot shows the distribution of genes in each set that are positively (red) or negatively (blue) correlated with Derlin-1 deficiency. (C) Representative immunofluorescence images with Ki67 (green), Tbr2 (red), and Hoechst (gray) staining of the DG in postnatal day 14 (P14) *Derl1*^{f/f} and *Derl1*^{NesCre} mice. Scale bars: 100 µm. (D–F) Quantification of the numbers of Ki67^+ cells (D) and Tbr2^+ cells (E) in the DG as well as the number of Tbr2^+ cells (F) in the molecular layer (ML) of the DG ($n = 3$ mice). (G) The percentage of Tbr2^+ cells in ML among total Tbr2^+ cells in the DG of P14 *Derl1*^{f/f} and *Derl1*^{NesCre} mice ($n = 3$ mice). (H) Experimental scheme for investigating the cell proliferation in the DG of P14 *Derl1*^{f/f} and *Derl1*^{NesCre} mice. (I) Representative immunofluorescence images with BrdU (red) and Hoechst (blue) staining of the DG in P14 *Derl1*^{f/f} and *Derl1*^{NesCre} mice. Scale bars: 100 µm. (J) Quantification of the numbers of BrdU^+ cells in the DG of P14 *Derl1*^{f/f} and *Derl1*^{NesCre} mice ($n = 3$ mice). (K) Comparison of P14 *Derl1*^{f/f} and *Derl1*^{NesCre} mice brain weights ($n = 3$ mice). (L) Representative immunofluorescence images with NeuN (green) and Hoechst (gray) staining of the cerebral cortex in P14 *Derl1*^{f/f} and *Derl1*^{NesCre} mice. Scale bars: 100 µm. (M) Experimental scheme for investigating the proliferation of NS/PCs and neurogenesis in *Derl2*^{f/f}, *Derl2*^{NesCre}, *Derl1*^{CaMKIIaCre(hetero)} (Control), and *Derl1*^{CaMKIIaCre} mice. (N) GSEA showing differential expression of 238 genes in the DG categorized by the GO term "Response to ER stress." GSEA shows gene expression changes in the DG of *Derl2*^{NesCre} mice relative to *Derl2*^{f/f} mice. The enrichment plot shows the distribution of genes in each set that are positively (red) or negatively (blue) correlated with Derlin-2 deficiency. (O–T) Representative immunofluorescence images of the DG with BrdU (red), DCX (cyan), and Hoechst staining (gray; insets) (O, R) and quantification of BrdU^+ proliferating cells (P, S) or BrdU^+ DCX^+ newborn immature neurons (Q, T) in mice of each genotype ($n = 3$ mice). Scale bars: 100 µm. (U) Representative immunofluorescence images with GFAP (green), S100β (red), Iba1 (cyan), and Hoechst (gray; insets) staining of the DG in 4-week-old *Derl1*^{f/f} and *Derl1*^{NesCre} mice. Scale bars: 100 µm. (V–Y) Quantification of the numbers of GFAP^+ S100β^+ astrocytes (V, X) and Iba1^+ microglia (W, Y) in the ML (V, W) or hilus (X, Y) of 4-week-old *Derl1*^{f/f} and *Derl1*^{NesCre} mice ($n = 3$ mice). (Z) Experimental scheme for assessing astrogenesis in the DG of *Derl1*^{f/f} and *Derl1*^{NesCre} mice. (AA) Representative immunofluorescence images of the DG with BrdU (red), S100β (green), and Hoechst staining (gray; insets). The areas outlined by a white rectangle are enlarged to the right. Scale bars, 100 µm (left images) and 20 µm (right images). The white arrows indicate merged cells. (AB) Quantification of the number of BrdU^+ S100β^+ newborn astrocytes in the SGZ and GCL of *Derl1*^{f/f} and *Derl1*^{NesCre} mice ($n = 3$ mice). (AC) Representative immunofluorescence images of the DG with active caspase3 (cyan), S100β (green), and Hoechst staining (gray; insets). The areas outlined by a white rectangle are enlarged to the right. Scale bars, 100 µm (left images) and 20 µm (right images). The white arrows indicate merged cells. (AD–AE) Quantification of the number of active caspase3^+ S100β^+ dying astrocytes in the SGZ (AD) and GCL (AE) of *Derl1*^{f/f} and *Derl1*^{NesCre} mice ($n = 3$ mice). (AF) Representative images of Nestin (green), GFAP (red), Sox2 (cyan), and Hoechst (gray; insets) staining of the DG in 2-month-old wild-type mice. (AG) Quantification of the number of radial Nestin^+ Sox2^+ and radial GFAP^+ Sox2^+ cells in the SGZ ($n = 3$ mice). Bar graphs are presented as the mean ± SEM. Significance was determined using Student's *t*-test (D–G, J, K, P, Q, S, T, V, W, X, Y, AB, AD, AE, AG). *$P < 0.05$ and **$P < 0.01$ determined by nominal GSEA *P* value (B, N). n.s. not significant.

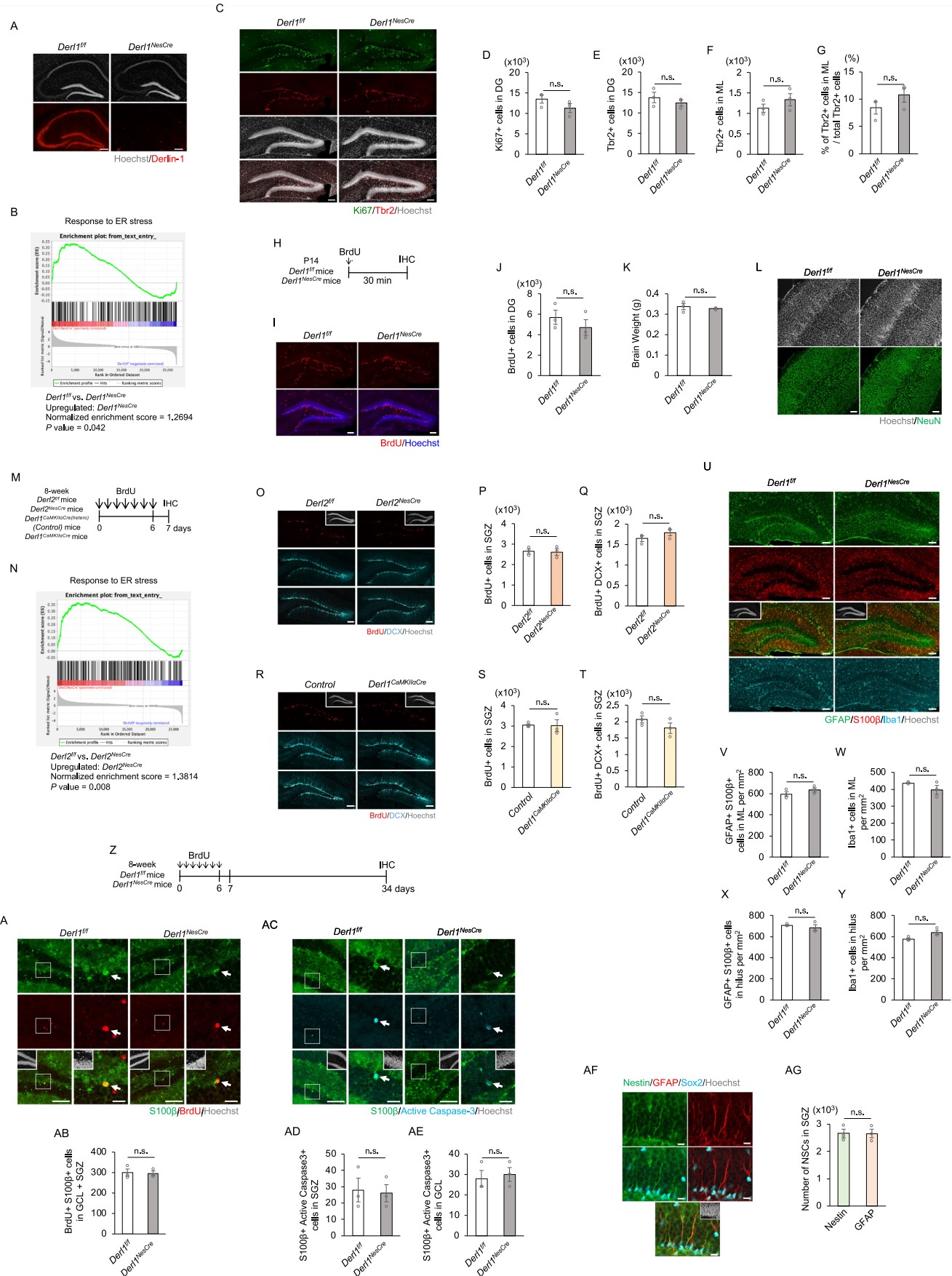

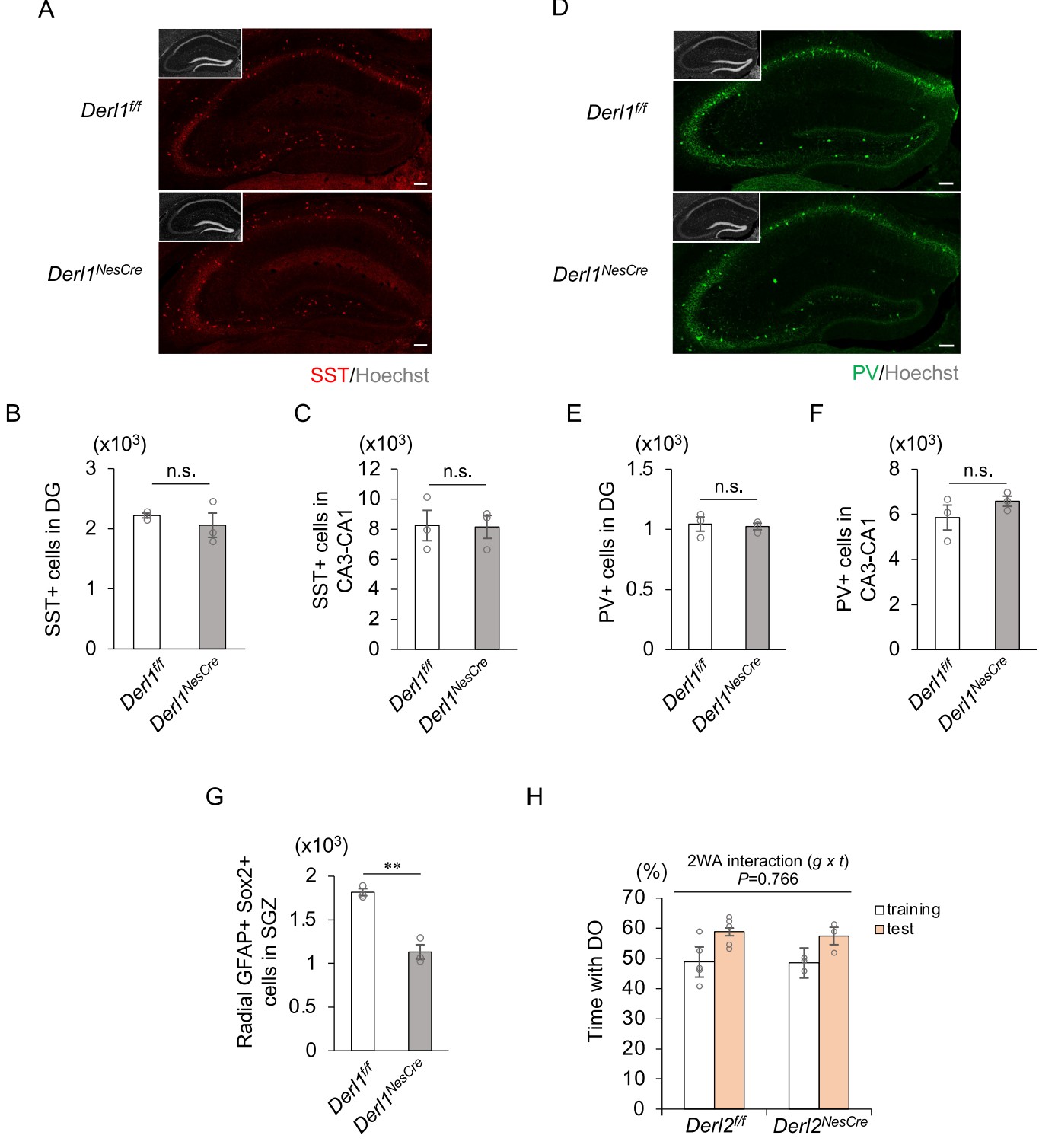

SST/Hoechst

PV/Hoechst

2WA interaction (g x t)
P=0.766

**Figure EV2.  Loss of *Derl1* does not alter the number of GABAergic interneurons in the hippocampus.**

(**A**) Representative immunofluorescence images of the hippocampus with somatostatin (SST) (red) and Hoechst staining (gray; insets). Scale bars: 100 μm. (**B, C**) Quantification of the number of SST$^+$ cells in the DG (**B**) and CA3-CA1 (**C**) regions of 2-month-old *Derl1*$^{f/f}$ and *Derl1*$^{NesCre}$ mice ($n = 3$ mice). (**D**) Representative immunofluorescence images of the hippocampus with parvalbumin (PV) (green) and Hoechst staining (gray; insets). Scale bars: 100 μm. (**E, F**) Quantification of the number of PV$^+$ cells in the DG (**E**) and CA3-CA1 (**F**) regions of 2-month-old *Derl1*$^{f/f}$ and *Derl1*$^{NesCre}$ mice ($n = 3$ mice). (**G**) Quantification of the number of radial GFAP$^+$ Sox2$^+$ NSCs in the SGZ of 4-month-old *Derl1*$^{f/f}$ and *Derl1*$^{NesCre}$ mice ($n = 3$ mice). (**H**) Percentage of time spent with the displaced object (DO) during the training and testing phase in 4-month-old *Derl2*$^{f/f}$ and *Derl2*$^{NesCre}$ mice ($n = 5$; *Derl2*$^{f/f}$ mice, $n = 3$; *Derl2*$^{NesCre}$ mice). 2WA two-way ANOVA, g genotype, t trial. Bar graphs are presented as the mean ± SEM. Significance was determined using Student's *t*-test (**B, C, E–G**) or two-way ANOVA (**H**). **$P < 0.01$ by Student's *t*-test (**G**). n.s. not significant.

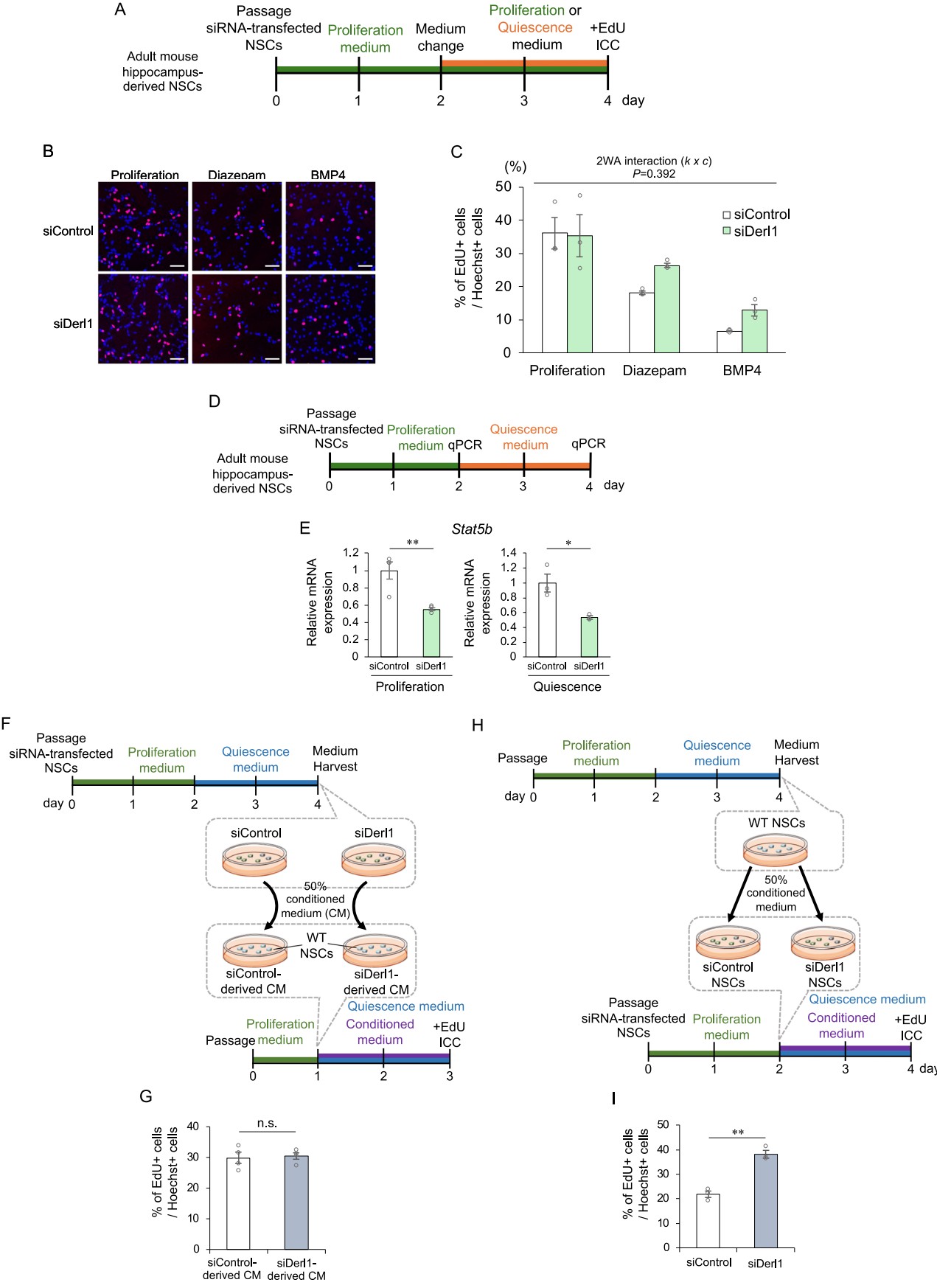

**Figure EV3. Inhibition of the transition from active to quiescent states in Derlin-1-deficient NSCs is cell-autonomously regulated.**

(A) Experimental scheme to induce the transition of control and *Derl1* knockdown mouse hippocampal NSCs from active to quiescent states. (B) Representative images of EdU (red) and Hoechst (blue) staining in siControl and siDerl1 mouse NSCs with or without induction of quiescence with diazepam (100 μM) or BMP4 (50 ng/mL) for 2 days. NSCs were fixed 30 min after the addition of EdU. Scale bars: 50 μm. (C) Quantification of the percentage of EdU$^+$ proliferating NSCs among total Hoechst$^+$ cells in siControl and siDerl1 mouse NSCs with proliferative conditions or induction of quiescence with diazepam or BMP4 for 2 days ($n = 3$ biological replicates). 2WA two-way ANOVA, k knockdown, c condition. (D) Experimental scheme for investigating the *Stat5b* expression of control and *Derl1* knockdown mouse NSCs from active to quiescent states. (E) Expression of *Stat5b* in siControl and siDerl1 mouse NSCs under proliferation and quiescent conditions. Gene expression levels were estimated by qPCR and normalized to that of *S18* ($n = 4$ biological replicates; Proliferation, $n = 3$ biological replicates; Quiescence). (F) Experimental scheme to investigate NSC proliferation with conditioned medium derived from control and *Derl1* knockdown NSCs over 2 days. (G) Quantification of the percentage of EdU$^+$ proliferating NSCs among total Hoechst$^+$ cells cultured for 2 days in siControl and siDerl1 NSC–derived conditioned quiescence medium ($n = 4$ biological replicates). (H) Experimental scheme to investigate NSC proliferation of control and *Derl1* knockdown NSCs with conditioned medium derived from wild-type (WT) NSCs over 2 days. (I) Quantification of the percentage of EdU$^+$ proliferating NSCs among total Hoechst$^+$ cells cultured for 2 days in siControl and siDerl1 NSCs with WT NSC–derived conditioned quiescence medium ($n = 3$ biological replicates). Bar graphs are presented as the mean ± SEM. Significance was determined using Student's *t*-test (E, G, I) or two-way ANOVA (C). *$P < 0.05$ and **$P < 0.01$ by Student's *t*-test (E, I). n.s. not significant.

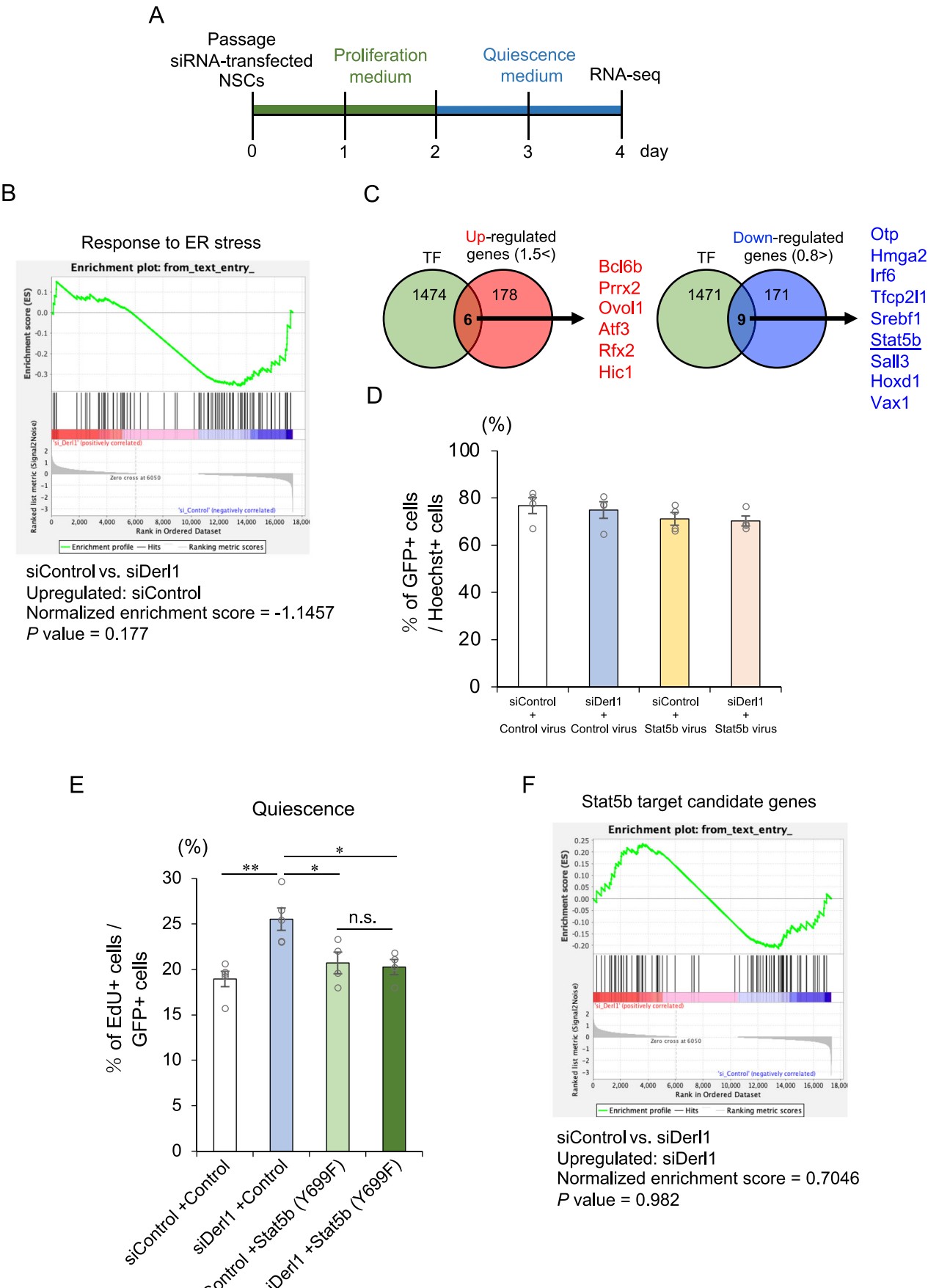

**A** Passage siRNA-transfected NSCs → Proliferation medium → Quiescence medium → RNA-seq
(days 0 1 2 3 4)

**B** Response to ER stress

siControl vs. siDerl1
Upregulated: siControl
Normalized enrichment score = -1.1457
*P* value = 0.177

**C** TF 1474 | 6 | Up-regulated genes (1.5<) 178 → Bcl6b, Prrx2, Ovol1, Atf3, Rfx2, Hic1

TF 1471 | 9 | Down-regulated genes (0.8>) 171 → Otp, Hmga2, Irf6, Tfcp2l1, Srebf1, Stat5b, Sall3, Hoxd1, Vax1

**D** % of GFP+ cells / Hoechst+ cells (%)
siControl + Control virus; siDerl1 + Control virus; siControl + Stat5b virus; siDerl1 + Stat5b virus

**E** Quiescence
% of EdU+ cells / GFP+ cells (%)
siControl +Control; siDerl1 +Control; siControl +Stat5b (Y699F); siDerl1 +Stat5b (Y699F)
** ; * ; * ; n.s.

**F** Stat5b target candidate genes

siControl vs. siDerl1
Upregulated: siDerl1
Normalized enrichment score = 0.7046
*P* value = 0.982

Figure EV4.   **Stat5b expression is decreased in Derlin-1-deficient NSCs, and the phosphorylation of Stat5b (Y699) is not required for the rescue of abnormal proliferation of Derlin-1-deficient NSCs.**

(A) Experimental scheme for investigating the molecular mechanism underlying the impairment of NSC transition to quiescence by *Derl1* knockdown. (B) GSEA showing differential expression of 92 genes in the NSCs categorized by the GO term "Response to ER stress." GSEA shows gene expression changes in siDerl1 NSCs relative to siControl NSCs. The enrichment plot shows the distribution of genes in each set that are positively (red) or negatively (blue) correlated with *Derl1* knockdown. (C) Venn diagrams showing the overlap between transcription factor (TF) genes and upregulated (left) or downregulated (right) genes in siDerl1 NSCs. (D) Quantification of the efficiency of each viral infection in siControl and siDerl1 NSCs ($n = 4$ biological replicates). (E) Quantification of the percentage of EdU$^+$ proliferating NSCs among total GFP$^+$ cells in siControl and siDerl1 NSCs with or without exogenous expression of mutant Stat5b (Y699F) [$n = 5$ biological replicates; + Control, $n = 4$ biological replicates; + Stat5b (Y699F)]. (F) GSEA showing differential expression of 80 candidate Stat5b target genes. GSEA shows gene expression changes in siDerl1 NSCs relative to siControl NSCs. The enrichment plot shows the distribution of genes in each set that are positively (red) or negatively (blue) correlated with *Derl1* knockdown. Bar graphs are presented as the mean ± SEM. Significance was determined using the nominal *P* value of GSEA software (B, F) or one-way ANOVA (E). *$P < 0.05$ and **$P < 0.01$ by one-way ANOVA followed by Bonferroni's post hoc test (E). n.s. not significant.

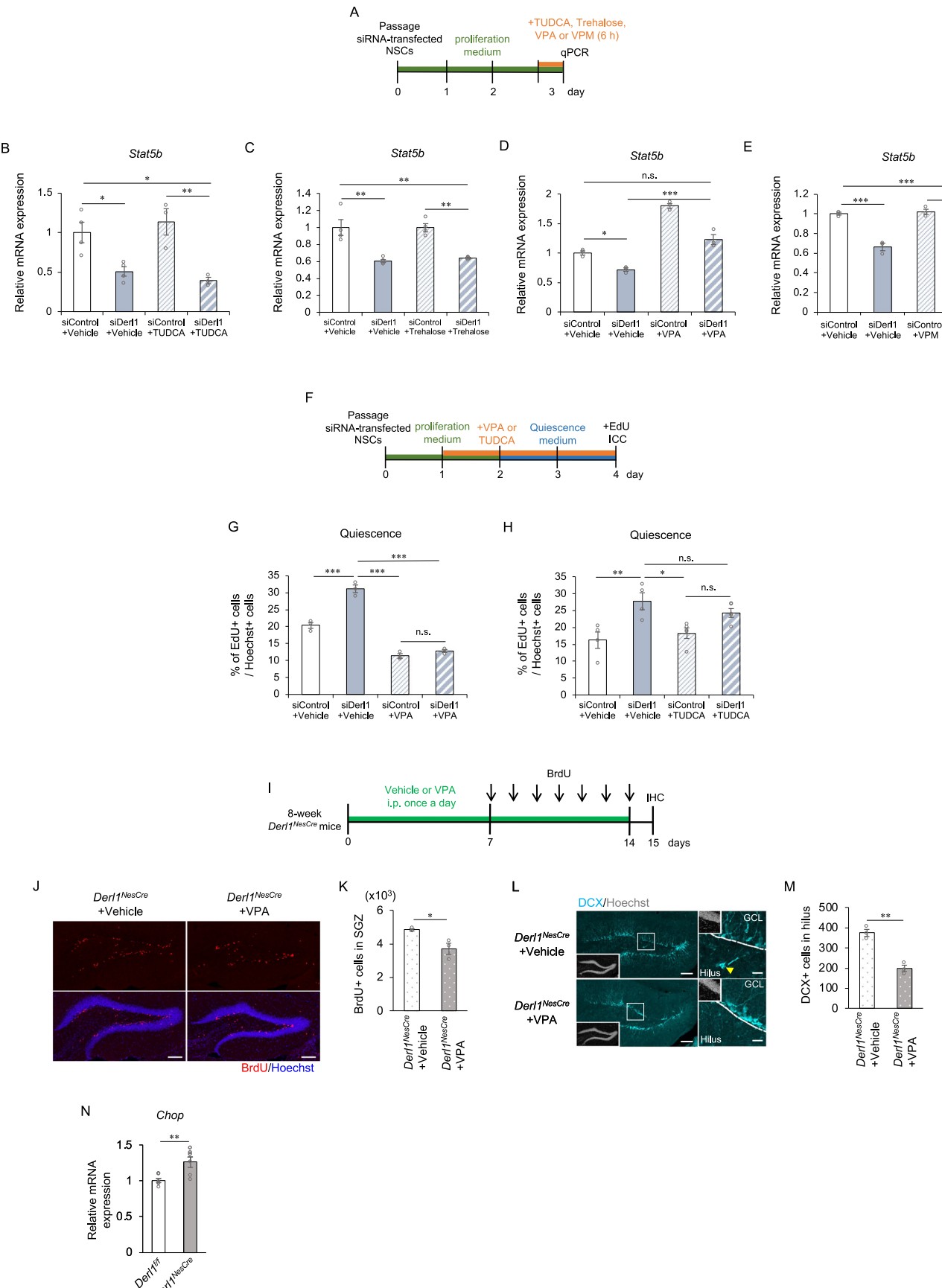

**Figure EV5. HDAC inhibitory activity, but not chaperone activity, increases *Stat5b* expression and inhibits the proliferation of NSCs.**

(A) Experimental scheme for assessing the expression of *Stat5b* in siControl and siDerl1 NSCs treated with or without TUDCA (50 μM), trehalose (10 mM), VPA (1 mM), or VPM (1 mM). (B–E) Expression of *Stat5b* in siControl and siDerl1 NSCs with or without TUDCA (B) ($n = 4$ biological replicates; Vehicle, $n = 3$ biological replicates; TUDCA), trehalose (C) ($n = 4$ biological replicates), VPA (D) ($n = 3$ biological replicates), or VPM (E) treatment ($n = 3$ biological replicates). Gene expression levels were estimated by qPCR and normalized to that of *β-actin*. (F) Experimental scheme for evaluating the effect of VPA (1 mM) or TUDCA (50 μM) on the impairment of the transition of NSCs to quiescence by *Derl1* knockdown. (G, H) Quantification of the percentage of EdU$^+$ proliferating NSCs among total Hoechst$^+$ cells in VPA-treated (G) ($n = 3$ biological replicates) or TUDCA-treated (H) ($n = 4$ biological replicates; Vehicle, $n = 5$ biological replicates; TUDCA) siControl and siDerl1 NSCs induced to enter the quiescent state by the administration of BMP4 for 2 days. (I) Experimental scheme for investigating the proliferation of NS/PCs in *Derl1*$^{NesCre}$ mice with or without VPA treatment. *Derl1*$^{NesCre}$ mice treated with vehicle or VPA daily for 2 weeks were simultaneously injected with BrdU daily for 7 days during the latter and fixed 1 day after the last BrdU injection. (J) Representative immunofluorescence images of the DG stained for BrdU (red) and Hoechst (blue) in *Derl1*$^{NesCre}$ mice treated with or without VPA. Scale bars: 100 μm. (K) Quantification of the number of BrdU$^+$ proliferating cells in the SGZ of *Derl1*$^{NesCre}$ mice treated with or without VPA ($n = 3$ mice). (L) Representative immunofluorescence images of the DG stained for DCX (cyan) and Hoechst (gray; insets) in *Derl1*$^{NesCre}$ mice treated with or without VPA. The areas outlined by a white rectangle are enlarged to the right. The yellow arrowhead indicates DCX$^+$ ectopic immature neurons in the hilus, and dashed white lines indicate the boundaries between the GCL and hilus. Scale bars, 100 μm (left images) and 20 μm (right images). (M) Quantification of the number of DCX$^+$ cells in the hilus in *Derl1*$^{NesCre}$ mice treated with or without VPA ($n = 3$ mice). (N) Expression of *Chop* in the DG of 2-month-old *Derl1*$^{f/f}$ and *Derl1*$^{NesCre}$ mice. Gene expression levels were estimated by qPCR and normalized to that of *S18* ($n = 7$; *Derl1*$^{f/f}$ mice, $n = 6$; *Derl1*$^{NesCre}$ mice). Bar graphs are presented as the mean ± SEM. *$P < 0.05$, **$P < 0.01$, and ***$P < 0.001$ by one-way ANOVA followed by Bonferroni's post hoc test (B–E, G, H) or Student's *t*-test (K, M, N). n.s. not significant.

