## [Peer Review File · EMBO Reports]

The Derlin-1-Stat5b Axis Maintains Homeostasis of Adult Hippocampal Neurogenesis

Naoya Murao, Taito Matsuda, Hisae Kadowaki, Yosuke Matsushita, Kousuke Tanimoto, Toyomasa Katagiri, Kinichi Nakashima, and Hideki Nishitoh

Corresponding author(s): Hideki Nishitoh (nishitoh@med.miyazaki-u.ac.jp) , Kinichi Nakashima (nakashima.kinichi.718@m.kyushu-u.ac.jp)

Review Timeline:

Submission Date:	26th Jul 23
Editorial Decision:	10th Oct 23
Revision Received:	5th Feb 24
Editorial Decision:	23rd Feb 24
Additional Correspondence:	13th Mar 24
Revision Received:	4th Jun 24
Editorial Decision:	24th Jun 24
Revision Received:	28th Jun 24
Accepted:	2nd Jul 24

Editor: Esther Schnapp

Transaction Report:

Dear Prof. Nishitoh,

Thank you for your patience while your manuscript was peer-reviewed at EMBO reports, and I apologize for our delayed response. Referee 3's report only came in last week. We have thus now received the full set of referee reports that is pasted below.

As you will see, the referees acknowledge that the findings are potentially interesting. However, together they raise the main concern that the study is not sufficiently thoroughly performed to allow for strong conclusions to be drawn. Referees 2 and 3 ask for a number of experiments to strengthen the study. We would need to ask you to perform major and significant revisions in order to proceed with your manuscript here. This is a borderline case, and I would like to know what kind of revisions you are willing and able to perform. We can discuss the exact revision requirements in a video chat, or by email, as you like.

In case we will agree on a set of revisions, I provide further information on manuscript revisions below:

I would like to invite you to revise your manuscript with the understanding that the referee concerns must be fully addressed and their suggestions taken on board. Please address all referee concerns in a complete point-by-point response. Acceptance of the manuscript will depend on a positive outcome of a second round of review. It is EMBO reports policy to allow a single round of major revision only and acceptance or rejection of the manuscript will therefore depend on the completeness of your responses included in the next, final version of the manuscript.

We realize that it is difficult to revise to a specific deadline. In the interest of protecting the conceptual advance provided by the work, we recommend a revision within 3 months (10th Jan 2024). Please discuss the revision progress ahead of this time with the editor if you require more time to complete the revisions.

- 1) A data availability section providing access to data deposited in public databases is missing. If you have not deposited any data, please add a sentence to the data availability section that explains that.
- 2) Your manuscript contains statistics and error bars based on $n=2$. Please use scatter blots in these cases. No statistics should be calculated if $n=2$.

5) a complete author checklist, which you can download from our author guidelines <https://www.embopress.org/page/journal/14693178/authorguide>. Please insert information in the checklist that is also reflected in the manuscript. The completed author checklist will also be part of the RPF.

6) Please note that all corresponding authors are required to supply an ORCID ID for their name upon submission of a revised

manuscript (<<https://orcid.org/>>). Please find instructions on how to link your ORCID ID to your account in our manuscript tracking system in our Author guidelines <<https://www.embopress.org/page/journal/14693178/authorguide#authorshipguidelines>>

I look forward to seeing a revised form of your manuscript when it is ready.

Kind regards,
Esther

Referee #1:

The present study investigates the role of ubiquitously expressed Derlin family members (Derlin-1, Derlin-2), components of the ER associated degradation pathway, in hippocampal neurogenesis. While previous reports implicate Derlin-1 or Derlin-2 in postnatal brain development, particularly of the cerebellum and striatum, whether Derlin-1 or Derlin-2 have roles in hippocampal neurogenesis is not known. Using Nestin-Cre to delete Derlin-1 or Derlin-2 postnatally, similar to their previous work (Sugiyama et al., iScience 2021), the authors show Derlin-1 is required for the transition of NSCs from active to quiescent stages and the location and survival of newborn neurons. Furthermore, loss of Derlin-1 increases seizure susceptibility and impairs cognitive function. Surprisingly, the activity of Derlin-1 does not appear to be through ER stress, but involve regulation of genes involved in stem cell states, such as HDAC inhibition. Overall, the study is compelling and the data are solid.

Referee #2:

This work by Murao et al investigated the role of Derl1, an ER-related molecule, in adult hippocampal neurogenesis. They found that Derl1 deficiency leads to abnormal proliferation of NSCs, ectopic localization and survival of newborn neurons, and disturbance of quiescent/active states of NSCs. They further showed a correlation between ectopic localization of newborn neurons and the increased seizure susceptibility in Derl1 deficient mice. Mechanistically, Stat5b expressed downstream of Derlin-1 regulates the transition of NSCs from active to quiescent states. Notably, they demonstrated that 4-PBA treatment can ameliorate abnormal hippocampal neurogenesis and the related behavioral deficits, as well as increased stat5b expression in NSCs. Some findings in this article are potential interesting, however, the manuscript is quite preliminary in its current form. There are some concerns but expanded mechanistic delineation are required.

Below, the major and minor points were outlined:

Major points:

1. Although the authors have excluded the involvement of neuronal Derl1 in adult neurogenesis, Cre recombinase driven by the Nestin promoter also deletes the target genes flanked by loxP sites in other cell types in the brain. In order to explore the specific role of Derl1 in adult neurogenesis, a tamoxifen inducible CreER driven by the Nestin promoter should be used for a temporal control of Derl1 inactivation without disrupting the early development of brain.
2. The ectopic localization of granular cells in the hilus was observed in Derl1-deficient mice. Whether Derl1-deficient mice display a spontaneous seizure? A multi-channel electroencephalogram (EEG) recording should be performed. This could explain whether spontaneous seizure results in ectopic localization of granular cells or ectopic localization of granular cells enhances seizure susceptibility in Derl1-deficient mice?
3. What is the molecular mechanisms mediating the survival of newborn neurons by Derl1?
4. I have mostly appreciated the experimental design and findings from in vitro analyses. However, does Derl1 knockout affect NSC activation, maintenance and/or reentry into quiescence from activation in vivo? The authors need to perform a more detailed analysis of the potential NSC phenotype.
5. Although the authors showed the correction of 4-PBA treatment and Stat5b expression in NSCs, they have not further proved whether the rescue effect of 4-PBA treatment on Derl1-deficient NSCs is directly through mediating Stat5b expression?
6. The authors argued that 4-PBA acts as HDAC inhibitor to rescue abnormal neurogenesis, seizure susceptibility and cognition. 4-PBA is a low-molecular-weight fatty acid and a nontoxic pharmacological compound that was originally used as a nitrogen-scavenging medication for chronic management of urea cycle disorders. A more specific HDAC inhibitor, such as VPA, should be used to strengthen the authors' claim.

Minor

1. In Fig3B, a panel of NSCs treated with proliferation medium should be illustrated.
2. The correlation between the increased seizure susceptibility and ectopically localized hilus neurons should be demonstrated. Are the ectopic Prox1+ neurons in the hilus of DerlNesCre mice more excitable? Considering the fact that Derlin-1 is depleted in the entire brain in DerlNesCre mice, how to exclude the possibility that other brain regions rather than hippocampus contributes to the enhanced seizure susceptibility of DerlNesCre mice?

Referee #3:

Title: The Derlin-1-Stat5b Axis Regulates the Homeostasis of Adult Hippocampal Neurogenesis.

The authors describe the role of Derl-1-Stat5 axis in the hippocampal neurogenic niche. Using an inducible transgenic mouse in which Derlin-1 expression is blocked in nestin-expressing cells (mostly NSCs).

This manipulation induces the hyperactivation of NSCs and disrupts the neurogenic process by generating neuronal progenitors with ectopic location... In parallel, Derlin1-deficient mice have higher susceptibility to seizures and perform worse in certain cognitive deficit tasks. Stat5 is proposed as responsible for the defects in neurogenesis in the derlin1-deficient mice. The overexpression of Stat 5 restores neurogenesis, and improves the susceptibility to seizures and cognitive dysfunctions. Finally, the authors test the potential restorative effects of 4-PBA in the derlin1-deficient mice.

The study is interesting and proposes a new mechanism controlling the dynamics of activation-quiescence of NSCs which in turn determine the neurogenetic output as NSC activation seems to be linked to their depletion.

A series of readouts are used throughout the work: The number of cell types and their proliferative rate and behavior. The relations among the quantification of cell types are used to suggest biological events. However, the inconsistency among experiments prevents such claims from being convincing.

NSCs are quantified in one experiment by nestin expression alone, then by GFAP and Sox2 in others. Real activation (very short-pulse of BrdU or Ki67) is not used thoroughly to measure NSC activation. The longer BrdU-pulses cannot be used for this. Also, NSCs cannot be distinguished by nestin alone or GFAP plus Sox2. GFAP and Sox2 are also expressed by all astrocytes in the DG. Nestin alone only labels filaments that cannot be readily attributed to a single NSC. To identify NSCs the slices must be co-labeled with nestin or vimentin or GFAP plus Sox2 to distinguish soma and nucleus and S100b, which labels astrocytes but not NSCs. The chosen combination should be used in all experiments the same.

Tbr2 is used as a marker of IPCs but it also labels NSCs and does not label all of the IPC. DCX labels immature neurons but also IPCs...the lineage study is not convincing and better use of markers in combination with carefully designed BrdU pulse-and-chase experiments being it is the gold standard for this type of analysis.

There is also a lot of confusion in the cell quantification about the area chosen, hilus, GCL, DG...For instance, there is a marked increase of DCX+ cells in the DG. Then no differences are found in the GCL but are found in the hilus. However, the increase of DCX+ cells in the hilus does not seem to account for the overall increase ("in the DG") by far.

NSC activation and differentiation and survival are processes at the core of the effects of Derlin1 deletion. They must therefore be addressed fully to be convincing. Cell death must be assessed as well as astroglialogenesis, and not only neurogenesis. It must be crystal clear when total numbers are used (and how they were obtained) versus proportions, and what is defined as the region of interest for the quantifications (hilus, GCL, ML, subgranular zone, ...).

Higher magnification pictures of co-localization studies are also required to demonstrate the appropriateness of the methodology.

All these quantifications, performed properly must be systematically run in all experiments (derlin1 deficiency, aging, rescue with stat5b...) other way there is no way to make a compelling mechanistic case.

Something that is striking is that the number of counted cells is very small. Following the explanation in the section methods, they quantify the number of cells and then they multiply by 6 and by 6 again. A clear example of this is the quantification at 9 months or 4 months, in the first one the number of Sox2+GFAP+ cells is 30 and 50 respectively, (which is very small compared to what is consistently shown in the bibliography). Also, the authors report a 20% of activation of NSCs, which is extraordinarily high.

The same goes for behavioral studies, they must be done the same in all cases. For instance, in the aging experiments, the author quantifies the number of NSCs at 9 months, however, the behavior is performed in 4-month-old mice.

Also, the in vitro studies suggest that the effects of derlin are cell autonomous. That the viral vectors to express stat5b transduce NSCs in enough numbers must be shown clearly. This would not rule out indirect effects in vivo as other cell types will also be affected, but at least that the viral vectors are able to act clearly on NSCs must be shown.

The in vitro studies are done with rats instead of with mice. Interference RNA is used to block the expression of derlin1, but using neurospheres derived from the inducible transgenic mice deficient in derlin1 to start with would have been a more systematic approach.

The problem is that the work is not systematic and often the lack of a rigorous approach subtracts strength from the claims. The manuscript is presented a bit like experiments somewhat unconnected experiments rather than follow a through and detailed experimental design.

Minor points:

The text must be revised to be more accurate because sometimes it lacks specificity. For instance: "Derlin-1 is expressed in adult hippocampal NSCs, and its expression 7 fluctuates across the stages of adult NSCs (Shin et al, 2015)." Explain better.

"acid (KA), an agonist for a subtype of ionotropic glutamate receptor, was administered to 2-month-old Der11NesCre and Der11f/f 16 control mice, and the seizure phenotype was observed for 1 h. Der11NesCre 17 mice showed higher seizure scores than control mice (Fig 2A and B)." How and what dose and how were seizures identified and measured?

The inclusion of the 4-APB is a bit puzzling because there is no direct link with the rest of the paper. Is surprising that based on the former effects reported for it, the authors find a fitting rescue of the abnormalities induced by the lack of derlin1. But without a thorough study of the same variables of the other experiments and more mechanistic insight, these results are merely anecdotal. I would definitely trade this section for a deeper, more rigorous, and detailed approach to the rest of the work.

As explained in the text, the main effect of derlin1 seems to be to promote NSC activation which translates into the acceleration of the natural depletion of NSCs. Yet the first manuscript to demonstrate the activation-linked depletion of NSCs and to propose that increased activation would lead to increased depletion is not cited (Encinas et al. 2011).

Statistics should be revised to include a 2/way ANOV analysis when two variables are involved.

Minor:

1- In the Figure 1E the author should include also the image of the Der11 f/f as in Der1 NesCre.

2- Figure 3B, there is no image of the proliferation in proliferative conditions, it would be better to include one to compare with the quiescence medium.

3- The Effect of quiescence medium in sicontrol neurospheres is even higher than in siDer11, around 20%. It is recommended to analyze using a 2-way ANOVA.

4- IT will be interesting to have the merge of the Figure 6B to understand the proliferative profile of the GCL, SGZ, and hilus....

5- In Figure 6C it would be interesting to have all the images in the same format, as Der11 NesCre+vehicle.

6- Why did the author analyze the expression of Tbr2 in the ML and not in Hilus (Figure 1 F)? Although Tr2 is a marker of IPCs, why does the author expect to find TBr2-positive cells in the ML?

Point-by-point responses are listed below. In the revised manuscript, the sections that have been revised or added in accordance with the comments are noted in red.

Arial: referees' comments.

Times new roman: authors' responses.

Referee #1:

The present study investigates the role of ubiquitously expressed Derlin family members (Derlin-1, Derlin-2), components of the ER associated degradation pathway, in hippocampal neurogenesis. While previous reports implicate Derlin-1 or Derlin-2 in postnatal brain development, particularly of the cerebellum and striatum, whether Derlin-1 or Derlin-2 have roles in hippocampal neurogenesis is not known. Using Nestin-Cre to delete Derlin-1 or Derlin-2 postnatally, similar to their previous work (Sugiyama et al., iScience 2021), the authors show Derlin-1 is required for the transition of NSCs from active to quiescent stages and the location and survival of newborn neurons. Furthermore, loss of Derlin-1 increases seizure susceptibility and impairs cognitive function. Surprisingly, the activity of Derlin-1 does not appear to be through ER stress, but involve regulation of genes involved in stem cell states, such as HDAC inhibition. Overall, the study is compelling and the data are solid.

We appreciate this reviewer's supportive comments on our manuscript.

Referee #2:

We thank the reviewer for the thoughtful comments. Additional experiments were conducted as indicated in the point-by-point responses below.

This work by Murao et al investigated the role of Der11, an ER-related molecule, in adult hippocampal neurogenesis. They found that Der11 deficiency leads to abnormal proliferation of NSCs, ectopic localization and survival of newborn neurons, and disturbance of quiescent/active states of NSCs. They further showed a correlation between ectopic localization of newborn neurons and the increased seizure susceptibility in Der11 deficient mice. Mechanistically, Stat5b expressed downstream of Derlin-1 regulates the transition of NSCs from active to quiescent states. Notably, they demonstrated that 4-PBA treatment can ameliorate abnormal hippocampal neurogenesis and the related behavioral deficits, as well as increased stat5b expression in NSCs. Some findings in this article are potential interesting, however, the manuscript is quite preliminary in its current form. There are some concerns but expanded mechanistic delineation are required.

Below, the major and minor points were outlined:

Major points:

1. Although the authors have excluded the involvement of neuronal Der11 in adult neurogenesis, Cre recombinase driven by the Nestin promoter also deletes the target genes flanked by loxP sites in other cell types in the brain. In order to explore the specific role of Der11 in adult neurogenesis, a tamoxifen inducible CreER driven by the Nestin promoter should be used for a temporal control of Der11 inactivation without disrupting the early development of brain.

Response: We understand the reviewer's concern that early developmental abnormalities in the brain may trigger abnormalities in adult neurogenesis. We therefore addressed this concern by examining the brain development of *Der11^{NesCre}* mice in as much detail as possible. Regarding the early development of *Der11^{NesCre}* mice, we previously reported no difference in brain size or weight at P0 (Sugiyama et al. *iScience* 2021, Reference Figure 1, below). The brain weight at P14 was also unchanged (revised Figure EV1K). Furthermore, we examined in detail by immunostaining for abnormalities in the early development of *Der11^{NesCre}* mice. Derlin-1 deficiency did not affect cell proliferation in the hippocampal dentate gyrus or cortical morphology at P14 (revised Figure EV1H–J, L). With these results and the finding that there was no clear difference in the number or localization of the neuronal progenitor marker Tbr2⁺ or Ki67⁺ cells in the hippocampus at P14 (Figure EV1C–G in the revised manuscript), Derlin-1 deficiency did not appear to have any apparent effect on brain development until at least 14 days of age. This point was described and carefully discussed in the revised manuscript (p5, lines 4–15). We quantified the number of glial cells to investigate the possibility that changes in cell types other than NSCs may affect adult neurogenesis. Immunostaining of astrocytes and microglia was performed in *Der11^{NesCre}* mice at 4 weeks of age, when the mice began to show abnormal proliferation of NSCs, but no clear difference was observed (revised Figure EV1U–Y).

These results suggest that *Der11^{NesCre}* mice do not exhibit conspicuous abnormalities in hippocampal development or glial cell development that directly affect adult neurogenesis. This point was described and carefully discussed in the revised manuscript (p6, lines 6-15). Of course, we fully agree with the reviewer's opinion that time-specific and cell type-specific gene loss due to inducible Cre expression is a direct experiment that addresses the reviewer's concerns. However, it would take several years to introduce these methods in our laboratory and analyze inducible Cre-expressing mice. The additional experiments described above did not reveal any abnormalities in brain development prior to P14, and we speculate that it is highly unlikely that the adult neurogenesis abnormalities in the *Der11^{NesCre}* mice were due to abnormalities in early brain development. These points, along with the use of other Derlin family knockout mice, will be addressed in more detail in future studies. We have discussed this matter with the editor and hope the reviewer agrees with our response.

Reference Figure 1 (Sugiyama et al. *iScience* 2021)

2. The ectopic localization of granular cells in the hilus was observed in *Der11*-deficient mice. Whether *Der11*-deficient mice display a spontaneous seizure? A multi-channel electroencephalogram (EEG) recording should be performed. This could explain whether spontaneous seizure results in ectopic localization of granular cells or ectopic localization of granular cells enhances seizure susceptibility in *Der11*-deficient mice?

Response: We understand the reviewer's concern. We have been observing *Der11^{NesCre}* mice for more than 10 years and have never observed spontaneous seizure symptoms except during kainic acid administration. We investigated whether *Der11^{NesCre}* mice exhibit spontaneous seizure symptoms in detail by continuously videotaping them for a longer period (21 h) in a stimulus-free, freely active state, and no seizure symptoms were observed in the *Der11^{NesCre}* mice (video data available in the Zenodo repository [<https://zenodo.org/records/10548867>]). This point was described in the revised manuscript (p7, lines 19-21). We believe that the presence or absence of spontaneous seizure symptoms during prolonged observation provides direct evidence. As the reviewer noted, a multichannel electroencephalogram (EEG) may capture minor changes in brain activity; we would like to address this point in more detail in the future, along with experiments in other Derlin family knockout mice, after setting up a physiological experimental system in our laboratory.

3. What is the molecular mechanisms mediating the survival of newborn neurons by *Der11*?

Response: Although there was no increase in the expression of ER stress response genes in cultured Derlin-1-deficient NSCs (Figure EV4B in the revised manuscript), the expression of ER stress response genes (Figure EV1B in the revised manuscript), including the ER stress-induced cell death-related gene *Chop* (revised Figure EV5N), was elevated in the dentate gyrus region of 2-month-old *Derl1^{NesCre}* mice. Thus, Derlin-1-mediated maintenance of ER homeostasis may be important for the survival of newborn neurons differentiated from adult NSCs in the brain. We discussed this point in the revised manuscript (p14, lines 1-6).

4. I have mostly appreciated the experimental design and findings from *in vitro* analyses. However, does *Derl1* knockout affect NSC activation, maintenance and/or reentry into quiescence from activation *in vivo*? The authors need to perform a more detailed analysis of the potential NSC phenotype.

Response: We understand the reviewer's concerns and are grateful for the thoughtful comments. We examined whether the phenotypes of NSCs inferred from *in vitro* experiments could be reconfirmed by *in vivo* experiments. The rate at which activated NSCs returned to the quiescent state was also significantly lower in *Derl1^{NesCre}* mice (revised Figure 1P–R), suggesting that Derlin-1 primarily regulates the quiescent and active states of NSCs. Furthermore, the number of activated NSCs in the *Derl1^{NesCre}* mice was still greater than that in the control mice after 3 weeks (revised Figure 1P and S), suggesting that activated Derlin-1-deficient NSCs are maintained over time. We included these data and discussed them in the revised manuscript (p6, line 33-p7, line 7).

5. Although the authors showed the correction of 4-PBA treatment and *Stat5b* expression in NSCs, they have not further proved whether the rescue effect of 4-PBA treatment on *Derl1*-deficient NSCs is directly through mediating *Stat5b* expression?

Response: We thank the reviewer for pointing this out. We examined whether the inhibitory effect of 4-PBA treatment on the transition to quiescence in Derlin-1-deficient NSCs was mediated by *Stat5b* expression through simultaneous knockdown of *Derl1* and *Stat5b* in 4-PBA-treated NSCs (revised Figure 5F). We observed that the ability of 4-PBA to rescue the impaired transition of Derlin-1-deficient NSCs from active to quiescent states was abrogated by *Stat5b* deficiency (revised Figure 5G). These results suggest that 4-PBA ameliorates the pathology of Derlin-1-deficient NSCs through *Stat5b* expression. We described these points in the revised manuscript (p11, lines 13-17).

6. The authors argued that 4-PBA acts as HDAC inhibitor to rescue abnormal neurogenesis, seizure susceptibility and cognition. 4-PBA is a low-molecular-weight fatty acid and a nontoxic pharmacological compound that was originally used as a nitrogen-scavenging medication for chronic management of urea cycle disorders. A more specific HDAC inhibitor, such as VPA, should be used to strengthen the authors' claim.

Response: We thank the reviewer for the suggestion and performed experiments to determine whether VPA, an

HDAC inhibitor, ameliorates abnormal adult neurogenesis in *Der11^{NesCre}* mice. VPA administration alleviated the aberrant proliferation of NSCs and the ectopic localization of immature neurons in *Der11^{NesCre}* mice, similar to the effects observed with 4-PBA treatment (revised Figure EV5I–M). These results suggest that HDAC inhibition contributes to the amelioration of abnormal adult neurogenesis in *Der11^{NesCre}* mice. These descriptions were included in the revised manuscript (p12, lines 2-8).

Minor

1. In Fig3B, a panel of NSCs treated with proliferation medium should be illustrated.

Response: Thank you for this comment. The data for NSCs treated with proliferation medium are shown (revised Figure 3B).

2. The correlation between the increased seizure susceptibility and ectopically localized hilus neurons should be demonstrated. Are the ectopic Prox1+ neurons in the hilus of *Der1NesCre* mice more excitable? Considering the fact that Derlin-1 is depleted in the entire brain in *Der1NesCre* mice, how to exclude the possibility that other brain regions rather than hippocampus contributes to the enhanced seizure susceptibility of *Der1NesCre* mice?

Response: We agree with the reviewer that increased seizure susceptibility may be caused by factors other than ectopic neurons. Therefore, the text was changed to weaken our claim (p7, lines 32-35). In addition, apart from ectopic neurons in the hilus, we quantified the number of interneurons in the hippocampal region that are known to be associated with increased seizure susceptibility in mice. The number of somatostatin⁺ or parvalbumin⁺ interneurons in the hippocampus of 2-month-old *Der11^{NesCre}* mice was comparable to that in the hippocampus of control mice, suggesting that at least hippocampal interneurons were less affected on seizure susceptibility (revised Figure EV2A–F). These descriptions were included in the revised manuscript (p7, lines 26-32).

Referee #3:

We thank the reviewer for the thoughtful comments. Additional experiments were conducted as discussed in the point-by-point responses below.

Title: The Derlin-1-Stat5b Axis Regulates the Homeostasis of Adult Hippocampal Neurogenesis.

The authors describe the role of Derl-1-Stat5 axis in the hippocampal neurogenic niche. Using an inducible transgenic mouse in which Derlin-1 expression is blocked in nestin-expressing cells (mostly NSCs).

This manipulation induces the hyperactivation of NSCs and disrupts the neurogenic process by generating neuronal progenitors with ectopic location... In parallel, Derlin1-deficient mice have higher susceptibility to seizures and perform worse in certain cognitive deficit tasks. Stat5 is proposed as responsible for the defects in neurogenesis in the derlin1-deficient mice. The overexpression of Stat 5 restores neurogenesis, and improves the susceptibility to seizures and cognitive dysfunctions. Finally, the authors test the potential restorative effects of 4-PBA in the derlin1-deficient mice.

The study is interesting and proposes a new mechanism controlling the dynamics of activation-quiescence of NSCs which in turn determine the neurogenetic output as NSC activation seems to be linked to their depletion.

A series of readouts are used throughout the work: The number of cell types and their proliferative rate and behavior. The relations among the quantification of cell types are used to suggest biological events. However, the inconsistency among experiments prevents such claims from being convincing.

NSCs are quantified in one experiment by nestin expression alone, then by GFAP and Sox2 in others. Real activation (very short-pulse of BrdU or Ki67) is not used thoroughly to measure NSC activation. The longer BrdU-pulses cannot be used for this. Also, NSCs cannot be distinguished by nestin alone or GFAP plus Sox2. GFAP and Sox2 are also expressed by all astrocytes in the DG. Nestin alone only labels filaments that cannot be readily attributed to a single NSC. To identify NSCs the slices must be co-labeled with nestin or vimentin or GFAP plus Sox2 to distinguish soma and nucleus and S100b, which labels astrocytes but not NSCs. The chosen combination should be used in all experiments the same.

Response: We thank the reviewer for the suggestion and agree with the importance of using consistent identification methods. Throughout the manuscript, we standardized the means of identifying NSCs by staining with GFAP and Sox2 antibodies, confirming the presence of Sox2⁺ cell bodies in the SGZ and the radial process of GFAP⁺ radial glial extension (radial GFAP⁺). This method of identifying NSCs using antibodies against GFAP and Sox2 is widely accepted and has been applied in many published studies (*e.g.*, Ehm O et al. *J Neurosci* 2010, Ashton RS et al. *Nat*

Neurosci 2012, Harris L et al. *Cell Stem Cell* 2021, Schneider et al. *EMBO J* 2022, Díaz-Moreno M et al. *PNAS* 2018). To further address the reviewer's concerns, we quantified the number of Sox2⁺ and Nestin⁺ (radial Nestin⁺) NSCs with radial glial extension and the number of Sox2⁺ and radial GFAP⁺ NSCs to determine whether there was a difference between the evaluation methods and found that they were comparable (revised Figure EV1AD and AE). We have carefully discussed this point in the revised manuscript (p6, lines 18-21).

Tbr2 is used as a marker of IPCs but it also labels NSCs and does not label all of the IPC. DCX labels immature neurons but also IPCs...the lineage study is not convincing and better use of markers in combination with carefully designed BrdU pulse-and-chase experiments being it is the gold standard for this type of analysis.

Response: We understand the reviewer's concern that it may not be appropriate to describe Tbr2⁺ cells as all IPCs and the need to more rigorously quantify each cell type (revised Figure 1N and O). Radial GFAP⁺, Sox2⁺ and Ki67⁺ cells were defined as active NSCs; Ki67⁺ cells minus radial GFAP⁺, Sox2⁺ cells as intermediate progenitor cells (IPCs); and DCX⁺ and Ki67⁻ cells, as immature neurons (revised Figure 1N). Interestingly, in contrast to what was reported in the previous version of our manuscript, the lineage progression index of immature neurons/IPCs was reduced in the *Derl1^{NesCre}* mice (revised Figure 1O). This finding was the result of redefining the cells based on the reviewer's comments. The reduced percentage of newly produced immature neurons compared to IPCs suggested that newborn neurons differentiated from Derlin1-deficient NSCs may not survive well in the SGZ of the dentate gyrus. This result is consistent with the findings of reduced cell viability in *Derl1^{NesCre}* mice (Figure 1M in the revised manuscript). We thank the reviewer for this comment and have carefully discussed this point in the revised manuscript (p6, lines 21-32).

There is also a lot of confusion in the cell quantification about the area chosen, hilus, GCL, DG...For instance, there is a marked increase of DCX⁺ cells in the DG. Then no differences are found in the GCL but are found in the hilus. However, the increase of DCX⁺ cells in the hilus does not seem to account for the overall increase ("in the DG") by far.

Response: We thank the reviewer for pointing this out and apologize for our confusing description. Figure 1D shows the number of BrdU and DCX double-positive cells in the DG, while Figure 1G shows the number of all DCX⁺ cells present in the SGZ. These results suggest that the number of DCX⁺ cells generated within a week, *i.e.*, newborn immature neurons, increases in the DG but that those DCX⁺ cells in the SGZ have low viability (revised Figure 1O). In other words, the majority of newborn neurons in the SGZ are not viable, while some ectopic newborn neurons in the hilus tend to survive, although the underlying mechanism is unknown. This expectation is based on the results showing the reduced survival of newborn neurons shown in Figure 1M in the revised manuscript. We have carefully discussed this point in the revised manuscript (p6, lines 27-32).

NSC activation and differentiation and survival are processes at the core of the effects of Derlin1 deletion.

They must therefore be addressed fully to be convincing. Cell death must be assessed as well as astrogliogenesis, and not only neurogenesis. It must be crystal clear when total numbers are used (and how they were obtained) versus proportions, and what is defined as the region of interest for the quantifications (hilus, GCL, ML, subgranular zone, ...).

Response: We fully agree with the reviewer's concern. Using a protocol similar to that shown in Figure 1J, we examined astrogliogenesis abnormalities in *Der11^{NesCre}* mice by S100 β staining 3 weeks after labeling with BrdU and measuring the number of newly generated BrdU⁺ and S100 β ⁺ astrocytes (revised Figure EV1Z and AA). In addition, the cell death of astrocytes in the DG was examined by active caspase-3 staining (revised Figure EV1AB and AC). The results confirmed that neither astrogliogenesis nor astrocyte cell death was altered by Derlin-1 deficiency. These descriptions were included in the revised manuscript (p6, lines 10-15).

Higher magnification pictures of co-localization studies are also required to demonstrate the appropriateness of the methodology.

Response: We thank the reviewer for pointing this out. Higher magnification images of the colocalization studies were added (revised Figure 1B, 1K and 2C).

All these quantifications, performed properly must be systematically run in all experiments (derlin1 deficiency, aging, rescue with stat5b...) other way there is no way to make a compelling mechanistic case.

Response: We agree with this comment and standardized the statistical analyses to ensure consistency throughout our manuscript.

Something that is striking is that the number of counted cells is very small. Following the explanation in the section methods, they quantify the number of cells and then they multiply by 6 and by 6 again. A clear example of this is the quantification at 9 months or 4 months, in the first one the number of Sox2+GFAP+ cells is 30 and 50 respectively, (which is very small compared to what is consistently shown in the bibliography).

Response: We apologize for our confusing description. In practice, a total of 10-12 brain slices collected every 6 slices were measured; the number of cells in all slices was summed and then multiplied by 6 to calculate the number of cells around the dentate gyrus, SGZ, and hilus within a single hippocampus. In other words, the multiplication by 6 was performed only once. We describe the calculation in detail in the methods section of the revised manuscript (p16, lines 28-31).

Also, the authors report a 20% of activation of NSCs, which is extraordinarily high.

Response: In the previous manuscript, we quantified NSCs only by staining with a nestin antibody, as the reviewer first commented. The number of active NSCs was re-quantified using GFAP, Sox2 and Ki67 antibodies in the revised manuscript. As a result, the percentage of activated NSCs was approximately 10% in the control mice (revised Figure 1O). We used the MCM2 antibody in the previous manuscript, but MCM2 is highly transcribed throughout the cell cycle, including in the G1 phase (Braun and Breeden, 2007; Tsuruga et al., 1997; Young and Tye, 1997), unlike Ki67 (Miller et al., 2018). Furthermore, the MCM2 protein is extremely stable, and its protein expression may be detected even after exit from the cell cycle (Harris L et al. *Cell Stem Cell* 2021). These differences may explain why the previous manuscript showed a greater percentage of active NSCs than did quantification using the Ki67 antibody. We thank the reviewer for this helpful remark.

The same goes for behavioral studies, they must be done the same in all cases. For instance, in the aging experiments, the author quantifies the number of NSCs at 9 months, however, the behavior is performed in 4-month-old mice.

Response: We fully agree with the reviewer's comment. We examined whether NSCs were already depleted in mice at the same age as those in which the novel location recognition test was performed. The number of NSCs was significantly decreased in 4-month-old *Der11^{NesCre}* mice (revised Figure EV2G). These descriptions were included in the revised manuscript (p8, lines 12-14).

Also, the in vitro studies suggest that the effects of darling are cell-autonomous. That the viral vectors to express stat5b transduce NSCs in enough numbers must be shown clearly. This would not rule out indirect effects in vivo as other cell types will also be affected, but at least that the viral vectors are able to act clearly on NSCs must be shown.

Response: We thank the reviewer for the suggestion. The efficiency of viral infection of NSCs was calculated as the ratio of the number of GFP⁺ cells to the number of total Hoechst⁺ cells, confirming that all the viruses used against siControl or siDer11 NSCs in the experiments infected more than 70% of the NSCs (revised Figure EV4D). These results suggest that the viral vectors used in our manuscript are able to act clearly on NSCs. We have added this description to the revised manuscript (p10 lines 7-9).

The in vitro studies are done with rats instead of with mice. Interference RNA is used to block the expression of derlin1, but using neurospheres derived from the inducible transgenic mice deficient in derlin1 to start with would have been a more systematic approach.

Response: We thank the reviewer for the suggestion. We obtained similar results with NSCs derived from *Der11^{NesCre}* mice (Reference Figure 2A–D, below) and with cultured rat NSCs (revised Figure 3C and D). Methods for artificially inducing quiescence in cultured rat NSCs are well established (Mira H et al. *Cell Stem Cell* 2010, Mukherjee S et al. *Nat Commun* 2016). Because lot-to-lot differences are often observed in primary cultured NSCs derived from

individual mice, especially in RNA-seq analysis, we decided to use well-established rat NSCs.

Reference Figure 2 (data not shown)

The problem is that the work is not systematic and often the lack of a rigorous approach subtracts strength from the claims. The manuscript is presented a bit like experiments somewhat unconnected experiments rather than follow a through and detailed experimental design.

Response: We are deeply grateful for the reviewer's insightful comments, which have demonstrably improved our revised manuscript. Their suggestions were invaluable.

Minor points:

The text must be revised to be more accurate because sometimes it lacks specificity. For instance: "Derlin-1 is expressed in adult hippocampal NSCs, and its expression 7 fluctuates across the stages of adult NSCs (Shin et al, 2015)." Explain better.

Response: We appreciate this comment and have added a more detailed explanation to the revised manuscript (p4, lines 8-10).

"acid (KA), an agonist for a subtype of ionotropic glutamate receptor, was administered to 2-month-old *Der11^{NesCre}* and *Der11^{f/f}* 16 control mice, and the seizure phenotype was observed for 1 h. *Der11^{NesCre}*

17 mice showed higher seizure scores than control mice (Fig 2A and B)." How and what dose and how were seizures identified and measured?

Response: We appreciate this comment and have added a more detailed explanation to the revised manuscript (p7, lines 21-25).

The inclusion of the 4-APB is a bit puzzling because there is no direct link with the rest of the paper. Is surprising that based on the former effects reported for it, the authors find a fitting rescue of the abnormalities induced by the lack of derlin1. But without a thorough study of the same variables of the other experiments and more mechanistic insight, these results are merely anecdotal. I would definitely trade this section for a deeper, more rigorous, and detailed approach to the rest of the work.

Response: We understand the concerns of the reviewer. We have attempted to elucidate the mechanisms upstream and downstream of Stat5b but have not succeeded at this time. Another important finding of our study is that 4-PBA can rescue the abnormal phenotype associated with adult neurogenesis. The experiments noted by reviewer #2 confirmed that 4-PBA ameliorates the impairment of Derlin-1-deficient NSCs through Stat5b expression (revised Figure 5F and G). Furthermore, the revised manuscript emphasizes our contention that 4-PBA-mediated phenotypic recovery of *Derl1*^{NesCre} mice is mediated by HDAC inhibitory activity, based on experiments in which mice were treated with VPA (revised Figure EV5I–M). We believe these points are significant in elucidating the mechanisms of brain dysfunction caused by disruption of ER quality control and in the search for treatments for these diseases.

As explained in the text, the main effect of derlin1 seems to be to promote NSC activation which translates into the acceleration of the natural depletion of NSCs. Yet the first manuscript to demonstrate the activation-linked depletion of NSCs and to propose that increased activation would lead to increased depletion is not cited (Encinas et al. 2011).

Response: We thank you for this comment and have cited this paper in the revised manuscript (p12, lines 34-36).

Statistics should be revised to include a 2/way ANOV analysis when two variables are involved.

Response: We understand the reviewer's concern and appreciate this comment. As a result of standardizing the way figures are presented throughout the manuscript, there are no longer any analysis that we believe require two-way ANOVA.

Minor:

1- In the Figure 1E the author should include also the image of the Derl1 f/f as in Derl NesCre.

Response: We appreciate the reviewers' comments and have addressed this suggestion (revised Figure 1E).

2- Figure 3B, there is no image of the proliferation in proliferative conditions, it would be better to include one to compare with the quiescence medium.

Response: We appreciate the reviewers' comments and have addressed this suggestion (revised Figure 3B).

3- The Effect of quiescence medium in sicontrol neurospheres is even higher than in siDer11, around 20%. It is recommended to analyze using a 2-way ANOVA.

Response: We thank the reviewers for their suggestions. As a result of the standardization of the presentation of the results throughout the manuscript, the graphs in Figure 3C (previous manuscript) were modified to be independent of each other (revised Figure 3C–E), and Student's t test was performed.

4- IT will be interesting to have the merge of the Figure 6B to understand the proliferative profile of the GCL, SGZ, and hilus....

Response: We appreciate the reviewers' comments and have addressed this suggestion by presenting a merged image of BrdU and Hoechst staining (revised Figure 6B).

5- In Figure 6C it would be interesting to have all the images in the same format, as Der11 NesCre+vehicle.

Response: We appreciate the reviewers' comments and have addressed this suggestion (revised Figure 6C).

6- Why did the author analyze the expression of Tbr2 in the ML and not in Hilus (Figure 1 F)? Although Tr2 is a marker of IPCs, why does the author expect to find TBr2-positive cells in the ML?

Response: We thank the reviewer for pointing this out and apologize for our confusing description. Figure EV1F and G show the results of experiments focused on hippocampal formation at P14. In the mouse DG and granular cell layer (GCL), the number of neuronal progenitor cells in the molecular layer (ML) decreases as DG development progresses (Noguchi et al. *J Neurosci* 2016). Therefore, to evaluate the hippocampal formation of *Der11^{NesCre}* mice at the developmental stage, we examined GCL morphology and the number and localization of Tbr2⁺ cells, a neuronal progenitor marker, at age P14 (Figure EV1C, E-G in the revised manuscript). To avoid misunderstanding, we corrected the text in the revised manuscript (p5, lines 2-11).

Dear Prof. Nishitoh,

Thank you for the submission of your revised manuscript. We have now received the comments from the referees who were asked to assess it, as well as cross-comments from referee 2.

As you will see, while referee 2 is initially positive, and both referees find your study interesting, referee 3 still has several concerns that all seem to be important. In the cross-comments, referee 2 agrees that these concerns should be addressed. I would thus like to give you the opportunity to do so, and I am also happy to discuss the exact revision requirements with you beforehand. You can, for example, send me a point-by-point response to the current concerns, which I can evaluate and we can discuss further, if necessary.

If we can agree on a new set of revisions, I am happy to invite you to revise your manuscript once more. Please let me know how you would like to proceed.

Referee #2:

The authors have addressed all the concerns.

Referee #3:

The authors describe the role of Derl-1-Stat5 axis in the hippocampal neurogenic niche. Using an inducible transgenic mouse in which Derlin-1 expression is blocked in nestin-expressing cells (mostly NSCs). The study is interesting and proposes a new mechanism controlling the dynamics of activation-quiescence of NSCs which in turn determine the neurogenetic output as NSC activation seems to be linked to their depletion.

We appreciate the efforts of the authors in the review the process. However, there are still important points that have not been resolved.

The author does not finish carrying out a consistent analysis of the quantifications; For example, the DCX-positive population is quantified only in the SGZ, however the DCX-positive population that proliferates is quantified in the DG (being an area that can not really be accurate for this purpose). The same occurs in the case of proliferation; in Figure 1C BrdU is measured in the DG while in Figure 4L in the SGZ, making the two measurements incomparable.

Although the author checked the expression of Nestin in NSCs, the number of NSCs that are quantified at all time points is extraordinarily low, which is far from all the literature published (see works by Kempermann; Encinas; Martín-Suárez...). Furthermore, the reduction in the numbers of NSCs that takes place between 2 and 4 months is even greater than that between 4 and 9 months. That is, according to the author's data, the loss of NSCs in two months (between 2 and 4 months) is greater than in 5 months (between 4 and 9), being the process of aging something totally absent in those quantifications.

In addition the author cannot make the claims of: 1- "The reduced percentage of newly produced immature neurons compared to IPCs suggested that newborn neurons differentiated from Derlin1-deficient NSCs may not survive well in the SGZ of the dentate gyrus" or 2- "In other words, the majority of newborn neurons in the SGZ are not viable, while some ectopic newborn neurons in the hilus tend to survive, although the underlying mechanism is unknown" since they are merely speculative and have no basis behind that statement.

Moreover, although the author checks the generation of astrocytes, (as it was asked for), 3 weeks is not an adequate time point, the most correct would be 4 at least. Furthermore, cell death is not quantified (nor in the GCL nor in the SGZ), which is a crucial point to understand the process itself.

I appreciate that the author has also performed new in vitro experiments, using mouse NSCs (Reference figure 2). The author draws the conclusion that both the rat cell line and the primary mouse cells behave the same. However, this is not entirely true, since in order to achieve the same effect between rat and mouse, the dose of diazepam in rats is 100uM while in mice it is

45uM. From this the author concludes that they do work the same, something that it is not really exact.

Also a major concern about the in vitro mouse NSCs cultures is the fact that the age selected to make the mouse cultures is P0. The use of P0 does not make sense to study the quiescence/activation capacity of NSCs since it behaves very differently in P0 and adults, (there are multiple protocols (Walter et al., 2011 for example) on culture of adult NSCs in vitro).

At the same time, the interpretation of the in vitro results must be reviewed. The author makes the claim that Stat5b or siDerl1 (Figure 4E and 5C respectively) produces an increase in the number of EdU cells but when this results are compared to the data obtained in the Figure 3C which is the proliferative condition the result changes. What really happens is that it is the control that reduces the proliferative rate, dropping to 10% in Figure 4E and 15% in Figure 4E. Therefore the interpretation of the results of 5C should be reconsidered.

There are some Minor point: Some graphs are cut on their Y axis, as is the example of Figure 1I, 1O, 2B, 2D....

In order not to perform a 2-way ANOVA, the author divides the graphs into two, which is not statistically correct since it omits the possibility of interaction from the analysis.

Cross-comments from referee 2:

I do agree with the referee 3, and his/her comments will definitely improve the manuscript. The authors may need to address several technical issues (the consistent analysis of quantification and in vitro cell model). Furthermore, the authors also need to tone down their claims: 1 "The reduced percentage of newly produced immature neurons compared to IPCs suggested that newborn neurons differentiated from Derlin1-deficient NSCs may not survive well in the SGZ of the dentate gyrus" or 2- "In other words, the majority of newborn neurons in the SGZ are not viable, while some ectopic newborn neurons in the hilus tend to survive, although the underlying mechanism is unknown". These claims could be appropriately made by lineage tracing using a transgenic mouse model that expresses a tamoxifen inducible creER driven by the nestin promoter. Unfortunately, the authors did not use this strategy in their initial study and in first round of revision.

Dear Hideki,

We have received comments from referee 2 on your point-by-point response now as well, which I paste below for your information.

Please address these last comments either experimentally or in the ms text or point-by-point response to the best of your abilities. Please also co-submit a final point-by-point response with your final ms.

I cannot judge the disagreement on NSC numbers and neither on the statistics. It would be good if you could discuss these issues in the ms text and the point-by-point response, and may be involve a statistics expert.

As a compromise, I think it is OK to us wildtype adult mouse NSC and knockdown *Derl1*.

I am looking forward to receiving your final ms file.

Referee 3:

My major concerns are:

1- The fact that the author just select some papers that justify their results with NSCs but omit the others that are different seems that the analysis and justification is biased, the effect of aging is something that is widely probed. Actually the second figure that they have used for justify is based on DCX cells and not NSCs, which is a little bit tricky because my concern as i explained in the review process is the number of NSCs, and the not effect of aging (something that is widely known) in these experiments.

2-Using the dentate gyrus as a measurement to evaluate a process that only takes place in the GCL more exactly in the SGZ is something that is not correct, although i explained it in the first revision they still maintain doing the analysis in the same way.

3- The author confirm that "it would be better to use hippocampus-derived adult NSCs from *Derl1^{NesCre}*" however as it "requires a lot of time and mice, and it may not be easy to establish" I think is not adequate to use d as "scientific" explanation to the reviewer. I believe that if you know that an experiment is better done in another model, taking into account that you know it, it should be done in the best experimental model although the effort was bigger. I think that EMBO journal is a high impact journal and this type of justification impacts on the quality of the journal.

4-The last comment is about the in vitro experiment and the proliferation. the author does not answer my concern which is the fact that:

What really happens is that it is the control , which should be the more stable condition,

reduces the proliferative rate up to 10% (compare figure 3C and 4E-5C), whereas the other conditions do not.

As a consequence the authors have an effect that is due to the fact that the control is not working properly and not due to the experimental condition or deletion

I checked the comment that you highlighted to me, that I should take into account before, sorry for that

My answer for that point is (in blue):

1-The author proposes this experiment that they consider more appropriate than using rats,

even so the most appropriate experiment is to use Der1NesCre mice since half of the paper (3 Figures of 6) are based on these in vitro experiments.

2-If I understood properly they used the same cells for the proliferation assay and then they add diazepam or BMP4, (the author said "*Adult rat hippocampal NSCs transfected with anti-Der1 siRNA were cultured for 2 days in proliferation medium, for another 2 days in proliferation medium or diazepam or BMP4-containing quiescence medium, and analyzed after 30 min of 5-ethynyl-2-deoxyuridine (EdU) treatment*"), then they should run a 2 way ANOVA because they measured the effect of the treatment and the proliferation rate. As a consequence the treatment can interact in the final results then they should use a 2 way ANOVA and not separate the experiment into a 3 different graph as they come from the same set of experiment.

Point-by-point responses are listed below. In the 2nd revised manuscript, sentences and figures that were revised or added in accordance with the comments are noted in red.

Arial: referees' comments.

Times New Roman: authors' responses.

Referee #3:

The authors describe the role of Derl-1-Stat5 axis in the hippocampal neurogenic niche. Using an inducible transgenic mouse in which Derlin-1 expression is blocked in nestin-expressing cells (mostly NSCs). The study is interesting and proposes a new mechanism controlling the dynamics of activation-quiescence of NSCs which in turn determine the neurogenetic output as NSC activation seems to be linked to their depletion.

We appreciate the efforts of the authors in the review the process. However, there are still important points that have not been resolved.

Comment-1

The author does not finish carrying out a consistent analysis of the quantifications; For example, the DCX-positive population is quantified only in the SGZ, however the DCX-positive population that proliferates is quantified in the DG (being an area that can not really be accurate for this purpose). The same occurs in the case of proliferation; in Figure 1C BrdU is measured in the DG while in Figure 4L in the SGZ, making the two measurements incomparable.

Response: We apologize for any inconsistency regarding the quantification of cell counts. As noted by the reviewer, all areas where the DCX-positive cells are counted are unified in the SGZ (revised Figure 1C, 1D, 2F, 6B, EV1P, EV1Q, EV1S, EV1T, and EV5K), except for the quantification of ectopic neurons (revised Figure 1F, 6C, and EV5M).

As a different point from the above, the NeuN-positive cells were counted in the GCL but described it as the DG. We corrected it to the GCL (revised Figure 1L and 1M).

Comment-2

Although the author checked the expression of Nestin in NSCs, the number of NSCs that are quantified at all time points is extraordinarily low, which is far from all the literature published

(see works by Kempermann; Encinas; Martín-Suárez...).

Response: We understand the reviewer's concern and agree that the number of NSCs we quantified is smaller than previously reported by other groups. Similar to our manuscript, a smaller number of NSCs is counted in some papers (Jessberger et al. *Exp Neurol* 2005, Bowers et al. *Cell Stem Cell* 2020), whereas the actual number of NSCs in the hippocampus of adult mice may be higher than our results, as the reviewer points out. Recognizing the importance of this point, we carefully read the methods in various papers and sought to understand why the quantitative results of the NSC varied from group to group. We found that there is a difference in whether or not the thickness of the brain sections used to count NSCs is taken into account. That is, while some groups measure NSCs taking into account the thickness of the brain sections, we and some other groups measured the number of NSCs in each brain section in a single plane. Furthermore, the results differ depending on whether the left and right hippocampal regions are quantified together or only one side. In our manuscript, unilateral hippocampal NSCs were counted according to several papers. We fully agree with that the number of NSCs in the hippocampus of adult mice is an important point as reviewer pointed out. We have measured NSCs using a uniform methodology throughout our manuscript and have carefully described the method, including the differences from previous papers (p16, line 34-p17, line 2, p17, lines 6-12) (p17, lines 7-12, p17, lines 15-22 in Merged PDF). It was also described that the number of NSCs in the adult mouse hippocampus varied among papers (p17, lines 6-12) (p17, lines 15-22 in Merged PDF). This response is also relevant to additional comment-1 below.

Comment-3

Furthermore, the reduction in the numbers of NSCs that takes place between 2 and 4 months is even greater than that between 4 and 9 months. That is, according to the author's data, the loss of NSCs in two months (between 2 and 4 months) is greater than in 5 months (between 4 and 9), being the process of aging something totally absent in those quantifications.

Response: We thank the reviewer for pointing this out. Quantification of NSCs in control and wild-type mice showed an average of 2654 (revised Figure EV1AG) at 2-month-old, 1963 (Figure 6G and EV2G in the revised manuscript) at 4-months-old, and 1292 (Figure 2D in the revised manuscript) at 9-month-old in our experiments. Converting the number of 2-month-old as 100%, this means that the number of 4-month-old decreased to 74.0% and 9-month-old to 48.7%. Guillemot group has examined in detail the changes in the number of NSCs with aging (Harris et al. *Cell Stem Cell* 2021). The number of NSCs in the DG is different from our results because of the reason that we described in above

response to the comment-2. We thus estimated the number of NSCs at 2-, 4-, and 9-month-old from their result of quantification of NSC numbers from 0.5- to 18-month-old and converted to percentages, and taking 2-month-old as 100%, the percentage decreased to 72.5% at 4-month-old and to 45.4% at 9-month-old. There is no big difference in the percentage reduction of NSCs between our results and those of Harris et al. In other words, the rate of reduction in the number of NSCs is likely to be greater between 2 and 4 months than between 4 and 9 months. This response is also relevant to additional comment-1 below.

Comment-4

In addition the author cannot make the claims of: 1- "The reduced percentage of newly produced immature neurons compared to IPCs suggested that newborn neurons differentiated from Derlin1-deficient NSCs may not survive well in the SGZ of the dentate gyrus" or 2- "In other words , the majority of newborn neurons in the SGZ are not viable, while some ectopic newborn neurons in the hilus tend to survive, although the underlying mechanism is unknown" since they are merely speculative and have no basis behind that statement.

Response: We fully agree with the reviewer that our assertion based on the quantitative results of DCX-positive cell counts is overstated. We have toned it down (p6, lines 25-26) (p6, lines 28-29 in Merged PDF).

Comment-5

Moreover, although the author checks the generation of astrocytes,(as it was asked for), 3 weeks is not an adequate time point, the most correct would be 4 at least. Furthermore, cell death is not quantified (nor in the GCL nor in the SGZ), which is a crucial point to understand the process itself.

Response: We agree with the reviewer's comment and examined astrocyte production in mice 4 weeks after the last BrdU injection (revised Figure EV1Z–AB). In addition, we quantified the number of astrocyte cell death in the GCL and the SGZ (revised Figure EV1AC–AE). Our results suggest that neither astrogliogenesis nor astrocyte cell death was altered by Derlin-1 deficiency. These descriptions were included in the revised manuscript (p6, lines 8-12) (p6, lines 10-15 in Merged PDF).

Comment-6

I appreciate that the author has also performed new in vitro experiments, using mouse NSCs (Reference figure 2). The author draws the conclusion that both the rat cell line and the

primary mouse cells behave the same. However, this is not entirely true, since in order to achieve the same effect between rat and mouse, the dose of diazepam in rats is 100uM while in mice it is 45uM. From this the author concludes that they do work the same, something that it is not really exact.

Also a major concern about the in vitro mouse NSCs cultures is the fact that the age selected to make the mouse cultures is P0. The use of P0 does not make sense to study the quiescence/activation capacity of NSCs since it behaves very differently in P0 and adults, (there are multiple protocols (Walter et al., 2011 for example) on culture of adult NSCs in vitro).

Response: We thank the reviewer for the thoughtful comments and agree that it is not appropriate to revalidate the results obtained by rat NSCs at different diazepam concentrations using P1 mouse NSCs. Methods for artificially inducing quiescence in cultured rat NSCs are well established (Mira et al. *Cell Stem Cell* 2010, Mukherjee et al. *Nat Commun* 2016), and our experimental results using rat NSCs may partially mimic the results *in vivo*. However, we understood the reviewer's concerns and therefore performed the experiments using adult mouse hippocampus-derived NSCs. Mouse NSCs also showed a trend toward a lower rate of return to quiescence with *Der11* knockdown using same dose of diazepam and BMP4 with rat NSCs (revised Figure EV3A–C). Furthermore, *Der11* knockdown reduced *Stat5b* expression in mouse NSCs as well as in rat NSCs (revised Figure EV3D and EV3E). We appreciate the reviewer's valuable comments and have incorporated these results into the revised manuscript (p8, lines 28-29, p9, lines 23-25) (p8, lines 32-33 in Merged PDF). This response is also relevant to additional comment-3 below. Please see also our detailed explanation in response to additional comment-3 below.

Comment-7

At the same time, the interpretation of the in vitro results must be reviewed. The author makes the claim that Stat5b or siDer11 (Figure 4E and 5C respectively) produces an increase in the number of EdU cells but when this results are compared to the data obtained in the Figure 3C which is the proliferative condition the result changes. What really happens is that it is the control that reduces the proliferative rate, dropping to 10% in Figure 4E and 15% in Figure 5C. Therefore the interpretation of the results of 5C should be reconsidered.

Response: We apologize for any misunderstanding that may have been caused by our confusing description. Figures 4E and 5C are the results of percentage of EdU-positive cells induced to enter a quiescent state using BMP4. The same conditions as the columns of siControl NSCs in Figures 4E and 5C are the columns of siControl NSCs treated with BMP4 in the revised Figure 3C (right white

column). We suspect that this misunderstanding might be caused by the separation of graphs into Figures 3C–E in the previously revised manuscript. We combined the three graphs into one as shown in revised Figure 3C (relevant to comment-8 below), and in revised Figures 4E and 5C, it is noted that the experiment was performed under quiescent conditions. In addition, since it may be difficult for readers to understand the culture conditions, the cell culture conditions are described above the graphs (revised Figure 4F, 4I, 5G, EV4E, EV5G, and EV5H). We hope these responses address the reviewer's concern. This is also relevant to additional comment-4 below.

Comment-8

There are some minor points: Some graphs are cut on their Y axis, as is the example of Figure 1I, 1O, 2B, 2D....

In order not to perform a 2-way ANOVA, the author divides the graphs into two, which is not statistically correct since it omits the possibility of interaction from the analysis.

Response: We apologize for our incorrect understanding of statistical analysis and thank the reviewer for his/her thoughtful comments. In order to perform a correct statistical analysis, we consulted with Dr. Nakai, an expert in biostatistics (<https://www.researchgate.net/scientific-contributions/Michikazu-Nakai-2007298944>). We revised our statistical analysis as follows. For Figures 3C–E, we analyzed by a 2-way ANOVA to verify the interaction of each factor as the reviewer pointed out. Since 2-way ANOVA revealed an interaction between each knockdown and each culture condition, a post hoc test was performed (revised Figure 3C). Furthermore, we changed the statistical analysis for the novel location recognition test to 2-way ANOVA (revised Figure 2H, 6H, and EV2H). We also changed the post hoc analysis after one-way ANOVA from the Tukey's test to the Bonferroni's test (revised Figure 4I, 5C, 5E, 5G, 6B, 6C, 6E, 6G, EV4E, EV5B–E, EV5G, and EV5H). This response is also addressed in the response to additional comment-6 below.

Cross-comments from referee 2:

I do agree with the referee 3, and his/her comments will definitely improve the manuscript. The authors may need to address several technical issues (the consistent analysis of quantification and in vitro cell model). Furthermore, the authors also need to tone down their claims: 1 "The reduced percentage of newly produced immature neurons compared to IPCs suggested that newborn neurons differentiated from Derlin1-deficient NSCs may not survive well in the SGZ of the dentate gyrus" or 2- "In other words, the majority of newborn neurons in the SGZ are not viable, while some ectopic newborn neurons in the hilus tend to survive,

although the underlying mechanism is unknown". These claims could be appropriately made by lineage tracing using a transgenic mouse model that expresses a tamoxifen inducible creER driven by the nestin promoter. Unfortunately, the authors did not used this strategy in their initial study and in first round of revision.

Response: We thank the reviewer for the thoughtful comments. In accordance with reviewer 3's comments, we revisited the quantification, etc. (see point-by-point above). We agree with the reviewer that our assertion from the quantitative results of DCX-positive cell counts is overstated. We have toned it down (p6, lines 25-26) (p6, lines 28-29 in Merged PDF). We fully agree with the reviewer's opinion that time-specific and cell type-specific gene loss due to inducible Cre expression is a direct experiment that addresses the reviewer's concerns. Since we do not have the tamoxifen-inducible CreER^{T2}-mice, we discussed about this point with editor and have taken action as previously commented with the editor's agreement. We recognize the importance of this point and hope that you will understand that this is an issue that we will address in more detail in the future.

Referee 3:

My major concerns are:

Additional comment-1

1- The fact that the author just select some papers that justify their results with NSCs but omit the others that are different seems that the analysis and justification is biased, the effect of aging is something that is widely probed. Actually the second figure that they have used for justify is based on DCX cells and not NSCs, which is a little bit tricky because my concern as i explained in the review process is the number of NSCs, and the not effect of aging (something that is widely known) in these experiments.

Response: We apologize for citing only papers that are consistent with our assertions. As noted in the response to comment-2 above, there may be several possible reasons why the number of NSCs varies from paper to paper. We have experimented with coherent quantitative methods throughout our manuscript and believe it is possible to discuss Derlin-1-dependent regulation of NSCs. We hope that the reviewer understand our response to comment-2 above.

Additional comment-2

2-Using the dentate gyrus as a measurement to evaluate a process that only takes place in the GCL more exactly in the SGZ is something that is not correct, although i explained it in the first revision they still maintain doing the analysis in the same way.

Response: As noted in the response to comment-1 above, we re-measured the number of cells in the appropriate location throughout manuscript. We deeply thank the reviewer for pointing this out.

Additional comment-3

3- The author confirm that "it would be better to use hippocampus-derived adult NSCs from *Der1^{NesCre}*" however as it "requires a lot of time and mice, and it may not be easy to establish" I think is not adequate to use d as "scientific" explanation to the reviewer.

I believe that if you know that an experiment is better done in another model, taking into account that you know it, it should be done in the best experimental model although the effort was bigger. I think that EMBO journal is a high impact journal and this type of justification impacts on the quality of the journal.

Response: We apologize for our misleading description. We attempted to culture NSCs from the hippocampal region of adult *Der11^{NesCre}* mice without success. At this time, we do not know why this is not possible using *Der11^{NesCre}* mice, even though NSCs can be cultured using wild-type mice. However, we understand the reviewer's concern and performed the experiments using wild-type adult mouse hippocampus-derived NSCs with the editor's agreement. Please see the response to comment-6 above for details. We hope that the reviewer understand our response.

Additional comment-4

4-The last comment is about the in vitro experiment and the proliferation. the author does not answer my concern which is the fact that:

What really happens is that it is the control , which should be the more stable condition, reduces the proliferative rate up to 10% (compare figure 3C and 4E-5C), whereas the other conditions do not.

As a consequence the authors have an effect that is due to the fact that the control is not working properly and not due to the experimental condition or delition

Response: We apologize for any misunderstanding caused by our confusing description. Please see our detailed description in response to comment-7 above. We hope that the reviewer understand our response.

I checked the comment that you highlighted to me, that i should take into account before, sorry for that

My answer for that 2 point is(in blue):

Additional comment-5

1-The author proposes this experiment that they consider more appropriate than using rats, even so the most appropriate experiment is yo use *Der11NesCre* mice since half of the paper (3 Figures of 6) are based on these in vitro experiments.

Response: We apologize for our confusing description. This point was addressed in our response to comment-6 and additional comment-3 above. We hope the reviewer agrees with our response.

Additional comment-6

2-If I understood properly they used the same cells for the proliferation assay and then they add diazepam or BMPO-4, (the author said "*Adult rat hippocampal NSCs transfected with anti-Der11 siRNA were cultured for 2 days in proliferation medium, for another 2 days in*

proliferation medium or diazepam or BMP4-containing quiescence medium, and analyzed after 30 min of 5-ethynyl-2-deoxyuridine (EdU) treatment), then they should run a 2 way ANOVA because they measured the effect of the treatment and the proliferation rate. As a consequence the treatment can intect in the final results then the should use a 2 way anova and not separate the experiment into a 3 different graph as they come from the same set of experiment.

Response: We apologize for our incorrect understanding of statistical analysis and thank the reviewer for pointing this out. We have integrated the graphs and changed the experiment to a 2-way ANOVA (revised Figure 3C). Please see our detailed description in response to comment-8 above.

Dear Prof. Nishitoh,

Thank you for the submission of your revised manuscript. We have now received the comments from referee 3, which are pasted below. Referee 3 still has one more suggestion that I would like you to address and incorporate before we can proceed with the official acceptance of your manuscript.

A few editorial requests will also need to be addressed:

- Please reduce the number of keywords to 5.
- Please note that the review tokens need to be removed and direct and accessible URLs added to the Data Availability Section (DAS) at the next stage.
- The Author Credits need to be removed from the ms file. All credits need to be entered during ms submission in our online system.
- Please move the Disclosure statement to after the Acknowledgements.
- The Funding information is not complete. The following info is missing in our online submission system: the Frontier Science Research Center, University of Miyazaki, Joint Usage and Joint Research Programs. Please add.
- All main and EV figures need to be uploaded as individual Figure files; the EV figure legends need to be added to the ms file, after the main figure legends.
- The following figures have two or more pages, but all figures need to fit on one single page: Figure 1, Figure 6, Figure EV1, Figure EV3, and Figure EV5. Please correct.
- The Appendix Table of Content needs page numbers added.
- The coloured text in the synopsis image is not easy to read. May be it can be made solid colour? I attach the image at its final size to this email for you to see.
- Please indicate the statistical test used for data analysis in the legends of figures EV1b, d-g, j-k, n, p-q, s-t, v-y, aa, ac, ae; EV 4b, f.
- Please note that information related to n is missing in the legends of figure 2b; EV 1k.
- Please note that the scale bar needs to be defined for figure 1n.
- Please note that the white arrows are not defined in the legend of figure 1n. This needs to be rectified.

I would like to suggest a few minor changes to the title and abstract. Please let me know whether you agree with the following:

The Derlin-1-Stat5b axis controls homeostasis of adult hippocampal neurogenesis

Adult neural stem cells (NSCs) in the hippocampal dentate gyrus continuously proliferate and generate new neurons throughout life. Although various functions of organelles are closely related to the regulation of adult neurogenesis, the role of endoplasmic reticulum (ER)-related molecules in this process remains largely unexplored. Here we show that Derlin-1, an ER-associated degradation component, spatiotemporally maintains adult hippocampal neurogenesis through a mechanism distinct from its established role as an ER quality controller. Derlin-1 deficiency in the mouse central nervous system leads to the ectopic localization of newborn neurons and impairs NSC transition from active to quiescent states, resulting in early depletion of hippocampal NSCs. As a result, Derlin-1-deficient mice exhibit phenotypes of increased seizure susceptibility and cognitive dysfunction. Reduced Stat5b expression is responsible for adult neurogenesis defects in Derlin-1-deficient NSCs. Inhibition of histone deacetylase activity effectively induces Stat5b expression and restores abnormal adult neurogenesis, resulting in improved seizure susceptibility and cognitive dysfunction in Derlin-1-deficient mice. Our findings indicate that the Derlin-1-Stat5b axis is indispensable for the homeostasis of adult hippocampal neurogenesis.

Best regards,

Esther

Referee #3:

The authors have responded to the comments adequately.

Despite this, there are still errors to be corrected, such as the axes of figures 1I, 1O, 2B, 6C, and 6D, EV1AG. The axes are cut off again, which reiterates the comments made in the previous review.

The error regarding the measurements also repeats, where the DG is again used as a measure. figure EV1D, E, J and EV2B, E.

Please correct or change or requantify the figures. Is the same error than the previous review...

Point-by-point responses are listed below.

Arial: referees' comments.

Times New Roman: authors' responses.

Referee #3:

The authors have responded to the comments adequately.

Response: We thank the reviewer for the thoughtful comments throughout the peer review.

Despite this, there are still errors to be corrected, such as the axes of figures 1I, 1O, 2B, 6C, and 6D, EV1AG. The axes are cut off again, which reiterates the comments made in the previous review.

Response: We apologize for our misunderstanding of your comments. We have corrected the y-axis not to cut off (revised Figure 1I, 1O, 2A, 2B, 6C, 6E, 6H, and EV1AG).

The error regarding the measurements also repeats, where the DG is again used as a measure. figure EV1D, E, J and EV2B, E.

Response: We thank the reviewer for pointing this out. In Figures EV1D, 1E, and 1J, we examined whether hippocampal development is affected by Derlin-1 deficiency using P14 mice. Some of neural progenitors which originally locate not only in the GCL or SGZ but also in the molecular layer during developmental stage and migrate into the GCL (Noguchi et al. *J Neurosci* 2016). To examine the effect of Derlin-1 deficiency in the hippocampal development, it is better to analyze the number of neural progenitors in the DG and the migration rate of neural progenitors from all of the DG to the GCL. We thus measure the number of neural progenitors in the DG. These points are described in p4, lines 1-10 in the revised manuscript (p4, lines 2-11 in Merged PDF). Figures EV2B and 2E investigate whether the involvement of GABAergic interneurons in seizure susceptibility in *Der11^{NesCre}* mice. Somatostatin- and parvalbumin-positive interneurons are present in all of the DG. As changes in their number may affect epileptic seizure susceptibility, their number was measured throughout the DG in this experiment. These points are described in p6, lines 19-25 in the revised manuscript (p6, lines 23-29 in Merged PDF).

Please correct or change or requantify the figures. Is the same error than the previous review...

Response: We apologize for the remaining issues raised in the previous review. We have responded as commented above and hope the reviewer will agree with our response.

Prof. Hideki Nishitoh
University of Miyazaki
medical Science
5200, Kihara, Kiyotake
Miyazaki 8891601
Japan

Dear Prof. Nishitoh,

I am very pleased to accept your manuscript for publication in the next available issue of EMBO reports. Thank you for your contribution to our journal.
